# RNA polymerase II stalling at pre-mRNA splice sites is enforced by ubiquitination of the catalytic subunit

Laura Milligan[1], Camille Sayou[1], Alex Tuck[2], Tatsiana Auchynnikava[1], Jane EA Reid[1], Ross Alexander[1†], Flavia de Lima Alves[1], Robin Allshire[1], Christos Spanos[1], Juri Rappsilber[1,3], Jean D Beggs[1], Grzegorz Kudla[4], David Tollervey[1]*

[1]Wellcome Trust Centre for Cell Biology, University of Edinburgh, Edinburgh, Scotland; [2]Friedrich Miescher Institute for Biomedical Research, Basel, Switzerland; [3]Institute of Biotechnology, Technische Universität Berlin, Berlin, Germany; [4]MRC Human Genetics Unit, Institute of Genetics and Molecular Medicine, University of Edinburgh, Edinburgh, Scotland

**Abstract** Numerous links exist between co-transcriptional RNA processing and the transcribing RNAPII. In particular, pre-mRNA splicing was reported to be associated with slowed RNAPII elongation. Here, we identify a site of ubiquitination (K1246) in the catalytic subunit of RNAPII close to the DNA entry path. Ubiquitination was increased in the absence of the Bre5-Ubp3 ubiquitin protease complex. Bre5 binds RNA in vivo, with a preference for exon 2 regions of intron-containing pre-mRNAs and poly(A) proximal sites. Ubiquitinated RNAPII showed similar enrichment. The absence of Bre5 led to impaired splicing and defects in RNAPII elongation in vivo on a splicing reporter construct. Strains expressing RNAPII with a $K_{1246}R$ mutation showed reduced co-transcriptional splicing. We propose that ubiquitination of RNAPII is induced by RNA processing events and linked to transcriptional pausing, which is released by Bre5-Ubp3 associated with the nascent transcript.

DOI: https://doi.org/10.7554/eLife.27082.001

*For correspondence:
d.tollervey@ed.ac.uk

Present address: †Institute of Life and Earth Sciences, Heriot-Watt University, Edinburgh, Scotland

Competing interests: The authors declare that no competing interests exist.

## Introduction

Over recent years, it has become increasingly clear that there are numerous, functionally important links between co-transcriptional RNA processing and the changes occurring on the transcribing RNA polymerase II (RNAPII). Most attention has focused on modifications of the regulatory C-terminal domain (CTD) of the large, catalytic subunit of RNAPII, which promote the ordered recruitment of numerous RNA packaging and processing factors to the nascent pre-mRNAs. However, there is also evidence that RNA processing events can affect the transcription machinery. In particular, it has been proposed that pre-mRNA splicing is associated with slowed RNAPII elongation in yeast (*Alexander et al., 2010b*; *Carrillo Oesterreich et al., 2010*), which presumably favors cotranscriptional splicing. In addition, an elongation 'checkpoint' exists upstream of the 3' cleavage and polyadenylation site in human cells (*Laitem et al., 2015*). These are predicted to reflect changes in the transcription machinery that are distinct from phosphorylation of the RNAPII CTD.

Analyses of the effects of UV-induced DNA damage revealed that RNAPII stalled at sites of DNA lesions undergo ubiquitination of the large polymerase subunit (termed Rpo21 or Rpb1 in yeast) (*Kvint et al., 2008*). RNAPII ubiquitination following DNA damage was reported to be a multi-step process, with initial mono-ubiquitination by Rsp5, followed by polyubiquitination by an Ecl1-Cul3 complex (*Harreman et al., 2009*). The stalled polymerase can be deubiquitylated by the ubiquitin

protease Ubp3, and failure of this activity leads to increased degradation of the polymerase by the proteasome. During the DNA damage response Ubp3 functions together with a cofactor Bre5 (*Bilsland et al., 2007*), with which it stably associates (*Li et al., 2005*). In addition, Ubp3 was identified as a factor physically associated with the transcription silencing factor Sir4 and loss of Ubp3 was reported to increase the efficiency of gene silencing near telomeres and at the inactive *HML* mating type locus (*Moazed and Johnson, 1996*). The Bre5-Ubp3 heterodimer also protects the TATA-binding protein Tbp1 against degradation (*Chew et al., 2010*), as well as functioning in a number of other, apparently unrelated pathways, including vesicle trafficking (*Baxter et al., 2005*; *Cohen et al., 2003*; *Ossareh-Nazari et al., 2010*) and starvation-induced autophagy of mature, cytoplasmic ribosomes (ribophagy) (*Kraft et al., 2008*). Possibly related to this, Bre5-Ubp3 also interacted genetically with the ribosome synthesis factor Nep1 (*Schilling et al., 2012*). Moreover, Bre5 was identified in a proteomic screen for proteins associated with poly(A)$^+$ RNA (*Mitchell et al., 2013*).

In order to identify new factors involved in RNA surveillance in yeast, genetic screens were performed, in which collections of strains individually deleted for each of the ~5000 non-essential genes were crossed *en masse* to strains lacking a single test gene (*Decourty et al., 2008*; *Milligan et al., 2008*). Following sporulation of the diploids generated, double mutant haploids were selected and allowed to grow for several generations. The mixed culture was then harvested, and the representation of each gene in the collection determined by microarray analysis. Unexpectedly, these competitive growth tests identified Bre5 and Ubp3 as showing the strongest, synergistic, negative growth defects with the non-essential, nuclear RNA surveillance factors Rrp47, which binds and activates the nuclease Rrp6, and the Air1 and Trf5 components of the TRAMP5 nuclear polyadenylation complex, which functions together with the exosome (*Decourty et al., 2008*; *Houseley and Tollervey, 2006*; *Losh et al., 2015*; *Milligan et al., 2008*; *Schuch et al., 2014*; *Stuparevic et al., 2013*).

The initial aim of this work was therefore to determine whether and how Bre5 and Ubp3 are linked to nuclear RNA metabolism.

## Results

### Bre5 is an RNA-binding protein that shows preferential association with exon 2 of spliced pre-mRNAs

In genome-wide synthetic lethal screens *bre5Δ* and *ubp3Δ* were each identified as showing strong synergistic negative growth with the exosome cofactors *rrp47Δ*, *air1Δ* and *trf5Δ* (*Decourty et al., 2008*; *Milligan et al., 2008*).

The crystal structure of Bre5 (*Li et al., 2005*) suggested the presence of a C-terminal RNA-recognition motif. The ability of Bre5, Ubp3 and the Bre5-Ubp3 dimer to bind RNA and single-stranded DNA was therefore tested in vitro (*Figure 1A*). Recombinant Bre5 and Ubp3 were expressed by in vitro translation singly or in combination using a TNT kit (Promega) and tested for binding to biotinylated oligos (A, A/U and dT) as previously described (*Milligan et al., 2008*). This showed that Bre5 bound to weakly to oligo(A), more strongly to oligo(U/A), and best to oligo(dT). Ubp3 alone showed little binding activity, but was recovered in the bound fraction when expressed together with Bre5. We concluded that Bre5 and the Bre5-Ubp3 complex can bind RNA and single-stranded DNA in vitro.

To confirm this, the ability of Bre5-Ubp3 purified from yeast to bind RNA was also tested. For this, Bre5 was expressed as fusion with a C-terminal, tripartite HTP tag (His6-TEV cleavage site-2x Protein A) under the control of the endogenous *BRE5* promoter. Bre5-HTP was purified by binding to an IgG column under native conditions (*Figure 1—figure supplement 1C*). As negative control a lysate from the wild type strain (BY4741) lacking the tagged form of Bre5 was also mock purified on IgG. As positive control the RNA-binding protein Nab3-HTP (*Holmes et al., 2015*) was purified and assessed for binding. Radiolabeled RNA oligonucleotides (oligos) were incubated with the IgG column and then washed extensively. Bound oligos were eluted, gel purified and quantified using a phosphor screen and Fuji scanner. The oligos used included the homopolymers, $A_{20}$, $U_{20}$, $G_{20}$ and $C_{20}$ (*Figure 1B* and *Figure 1—figure supplement 1A*). In addition, in vivo Bre5-binding sites identified by CRAC were analyzed to identify enriched motifs (see below). This identified the motif UUUG (*Figure 1—figure supplement 1B*) as the preferred in vivo target for Bre5. Oligos were generated

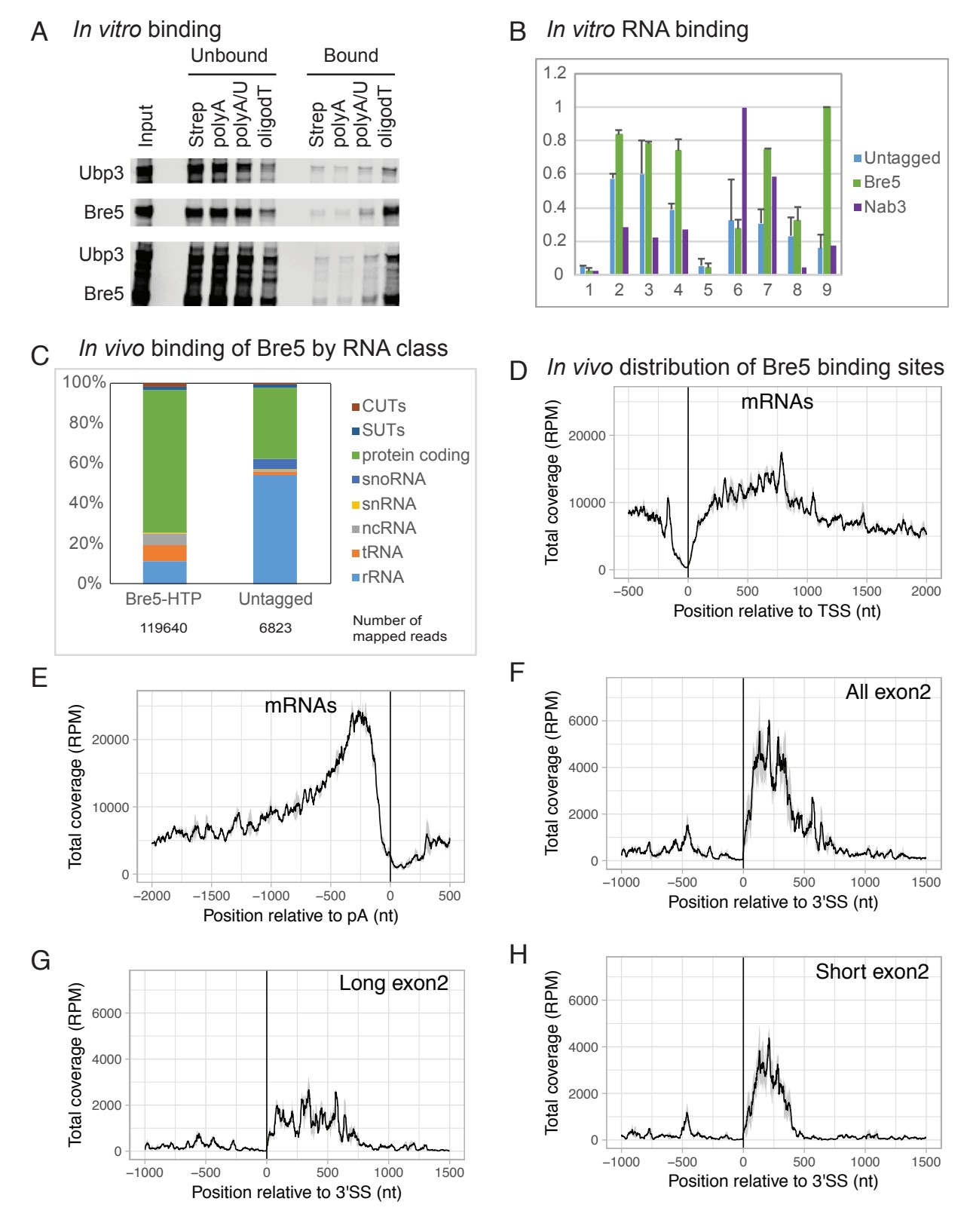

**Figure 1.** Bre5 binds RNA in vitro and in vivo. (**A**) Bre5 and Ubp3 plus Bre5 bind RNA in vitro. Recombinant Bre5 and Ubp3 were expressed in *E.coli* individually and in combination, and bound to the biotinylated oligonucleotides indicated, which were immobilized on streptavidin agarose columns.
*Figure 1 continued on next page*

*Figure 1 continued*

(B) Tagged Bre5 and Nab3 were purified from yeast, along with a mock-purified, untagged control, and assayed for binding to labeled RNA oligonucleotides. 1: polyA (AAAAAAAAAAAAAAAAAAAAAAAAAA),. 2: polyG (GGGGGGGGGGGAGGGGGGGGGGGAGGGGG). 3: polyU (UUUUUUUUUUUUUUUUUUUUUUUUUU),. 4: polyC (CCCCCCCCCCCACCCCCCCCCCCACCCCC),. 5: polyA/U (AUAUAUAUAUAUAUAUAUAUAUAUAU),. 6: Nab3-A (AAAAAUCUUAAAUCUUAAAUCUUAAAAA),. 7 Nab3-U (UUUUUUCUUUUUUCUUUUUUCUUUUUUU),. 8: Bre5-A (AAAAAUUUGAAAUUUGAAAUUUGAAAAA),. 9: Bre5-U (UUUUUUUUGUUUUUUGUUUUUUGUUUUU). The y axis represents the signal quantified from the bound oligos (arbitrary units), the error bars are the standard deviation from three biological replicates. See also *Figure 1—figure supplement 1A–C*. (C) Distribution of RNA sequences recovered with Bre5-HTP and the untagged control across different RNA classes. 15-fold fewer reads were recovered with the control (*Supplementary file 1*, Table S2) and numbers of mapped reads are indicated below the bar graphs. (D) Metagene analysis of the distribution of RNA sequences recovered with Bre5 across protein coding transcripts (5171 features) (*Xu et al., 2009*), in reads per million (RPM). Sequences were aligned relative to the transcription start site (TSS). The average of 2 Bre5 CRAC replicates is shown. The standard deviation appears as a grey shadow. (E) As D, but sequences were aligned relative to the poly(A) site (pA). (F) Metagene analysis of the distribution of RNA sequences recovered with Bre5 across intron containing, protein coding transcripts (288 features), in RPM. Sequences were aligned relative to the 3' splice site (3'SS). The average of 2 Bre5 CRAC replicates is shown. The standard deviation appears as a grey shadow. (G) As F, but data were filtered to show only genes with long exon2 regions (above 600 nt, 145 features). (H) As F, but data were filtered to show only genes with short exon2 regions (below 600 nt, 143 features).

DOI: https://doi.org/10.7554/eLife.27082.002

The following source data and figure supplements are available for figure 1:

**Source data 1.** Source data for *Figure 1B*.
DOI: https://doi.org/10.7554/eLife.27082.005

**Figure supplement 1.** Bre5 binding to RNA is independent of Ubp3 or the ubiquitin ligase Rsp5.
DOI: https://doi.org/10.7554/eLife.27082.003

**Figure supplement 2.** Distribution of Bre5 across individual genes.
DOI: https://doi.org/10.7554/eLife.27082.004

with three copies of UUUG, interspersed and flanked with either A residues (oligo Bre5-A) or U (oligo Bre5-U) (see legends to *Figure 1* and *Figure 1—figure supplement 1*). The consensus Nab3-binding site (UUCU) was used to generate similar oligos (Nab3-A and Nab3-U).

Relative to the column loaded with lysate from the untagged strain, the column loaded with Bre5-HTP recovered ~1.5 fold more oligo(A), two fold more oligo(U), but similar levels of oligo(G) and oligo(C) (*Figure 1B* and *Figure 1—figure supplement 1*). Strikingly, the strongest binding was observed for the Bre5-U oligo on the Bre5 column; six fold greater than the BY control. Substantial binding was also seen for the Nab3-U oligo, which differs in only three positions from Bre5-U (*Figure 1B*). Binding to the Nab3 column was significantly different from Bre5, with strong association of the Nab3-A and Nab3-U oligo, but low association with the Bre5 target oligos (*Figure 1B* and *Supplementary file 1*, Table S2). We conclude that Bre5 can bind RNA in vitro, with some sequence specificity.

To determine whether Bre5 also directly binds RNA in vivo, we used the approach of UV crosslinking and analysis of cDNA (CRAC) (*Granneman et al., 2009*). Actively growing cultures expressing Bre5-HTP were briefly UV-irradiated (100 s) and Bre5-HTP was recovered under stringent, denaturing conditions. Bound RNA was partially degraded, amplified by RT-PCR and identified by illumina sequencing. Mapping the reads to the yeast genome (*Figure 1C*) showed a predominance of RNAPII targets, particularly protein-coding genes, although hits in rRNA and tRNAs were also recovered at lower frequency. Compared to Bre5, the untagged control recovered an average of ~15 fold fewer total reads following PCR amplification. Numbers of mapped reads are indicated below the bar graphs in *Figure 1C*.

A metagene analysis of the distribution of Bre5 binding across all protein-coding genes was performed (*Figure 1D–G*). Alignment by the transcription start site (TSS) revealed marked depletion of Bre5 over the promoter proximal region of ~150 nt relative to sites further 3' (*Figure 1D*). In contrast, strong enrichment was seen for Bre5 binding ~200 nt upstream from the mRNA 3' cleavage and polyadenylation sites, followed by a sharp decrease (*Figure 1E*). Comparison of Bre5 binding on intron-containing, protein-coding genes aligned by the 3' splice site showed enrichment on the exon 2 regions relative to exon 1 or introns (*Figure 1F*). Since Bre5 binding is elevated close to the poly (A) site, it seemed possible that this was responsible for the apparent exon 2 signal. However, stratifying the data by exon 2 length revealed that the high signal at the 5' end of exon 2 persisted even on genes with exon 2 longer than 600nt (*Figure 1G*). Bre5 binding was most marked on genes with

relatively short exon 2 sequences (*Figure 1H*), on which transcription pausing is reported to be stronger (*Carrillo Oesterreich et al., 2010*). We also examined the distribution of Bre5 on individual genes (*Figure 1—figure supplement 2*). This showed the same trends as the metagene analysis.

To identify preferred targets for Bre5 binding in vivo, sequences generated by CRAC were analyzed to identify potential recognition motifs using MEME (see Experimental Procedures). Analyses of two independent, replica data sets each identified the motif UUUG as the strongest interaction target, present in 70% of target sequences analyzed (e > $10^{-120}$). As a positive control, CRAC data for the RNA-binding protein Nab3 (*Bresson et al., 2017*) were similarly analyzed and identified the previously reported UUCU-binding site. Analysis of negative control data from the BY control strain, Rpo21 (the large subunit of RNAPII) or the histone methyl-transferases Set1 and Set2 (*Sayou et al., 2017*), identified no clearly enriched motifs.

Bre5 forms a stable dimer with the ubiquitin protease Ubp3 (*Li et al., 2005*), suggesting that Bre5 might be recruited to nascent transcripts through binding of Ubp3 to ubiquitinated forms of RNAPII or other transcription factors. To assess this, crosslinking of Bre5-HTP was performed in a *ubp3Δ* strain. In the absence of Ubp3, we saw increased relative Bre5 binding close to the TSS, whereas binding in exon 2 and close to the poly(A) site was not clearly altered (*Figure 1—figure supplement 1D, E and F*). This observation indicates that the major pathway for Bre5 recruitment does not involve Ubp3 binding to ubiquitinated substrates. Rsp5 was previously reported to ubiquitinate Rpo21 (*Harreman et al., 2009*), so Bre5-HTP crosslinking was also tested in an *rsp5-3* temperature sensitive (ts) strain (*Neumann et al., 2003*) following transfer to non-permissive temperature (37°C). No clear differences were observed close to the TSS or poly(A) sites. A modest reduction was observed in Bre5 binding to RNAPII transcripts at the 5′ ends of exon 2 regions (*Figure 1—figure supplement 1G, H and I*). It is unclear whether this is a direct consequence of reduced activity of Rsp5, which is highly pleiotropic. We conclude that Bre5 can bind RNAPII transcripts independently of Ubp3, with elevated binding in exon 2 regions of spliced pre-mRNA and upstream of the 3′ cleavage and polyadenylation sites.

To assess binding of Bre5 to unspliced pre-mRNAs relative to splicing products, the number of reads spanning the spliced (exon-exon, EE) to unspliced junctions (exon-intron, EI and intron-exon, IE) was calculated as 2EE/(EI + IE) for each transcript and averaged between replicates (*Milligan et al., 2016*; *Tuck and Tollervey, 2013*). An 2EE/(EI + IE) ratio >1 indicates preferential binding after pre-mRNA splicing. For the large subunit of RNAPII, Rpo21, the 2EE/(EI + IE) ratio was 0.2, showing the expected association with unspliced nascent transcripts. For Bre5, the ratio was 30, indicating preferential binding to the spliced products (*Supplementary file 1*, Table S2). Notably, the 2EE/(EI + IE) ratio was further increased for Bre5 in the *ubp3Δ* strain, largely due to reduced intron binding, indicating that Upb3 does contribute to early pre-mRNA recruitment of Bre5. In contrast, no clear differences were seen in the Bre5 strain also carrying the ts-lethal *rsp5-3* mutation at either 23°C or 37°C.

We interpret the results in *Figure 1* and *Figure 1—figure supplement 2* and *Supplementary file 1*, Table S2, as indicating that Bre5-Ubp3 associates with the transcript following the successful completion of splicing.

## Loss of Bre5 results in RNAPII stalling and decreased mRNA on an inducible reporter

Inspection of the distribution of Bre5 hits over exon 2 of pre-mRNAs, showed that binding was highest on genes with short exon 2 regions (*Figure 1F,G and H*). Previous analyses have reported that pre-mRNA splicing is associated with transcription pausing on exon 2, which is more frequent on genes with short exon 2 regions, perhaps to allow time for co-transcriptional splicing (*Alexander et al., 2010b*; *Carrillo Oesterreich et al., 2010*).

To test for effects of Bre5 on transcription pausing by RNAPII we made use of 'RiboSys' reporters, which were developed to generate kinetic data for pre-mRNA transcription and splicing (*Alexander et al., 2010a*). The RIBO1 reporter (*Figure 2A*) contains part of the *PGK1* open reading frame with the insertion of the *ACT1* pre-mRNA intron, integrated into the chromosomal *HIS3* locus under the control of an inducible tetO7 promoter. Following induction by addition of doxycycline, samples were analyzed for pre-mRNA and mRNA levels by RT-PCR, and RNAPII occupancy by chromatin immunoprecipitation (ChIP) at five locations along the reporter (probes 1–5 in *Figure 2A*).

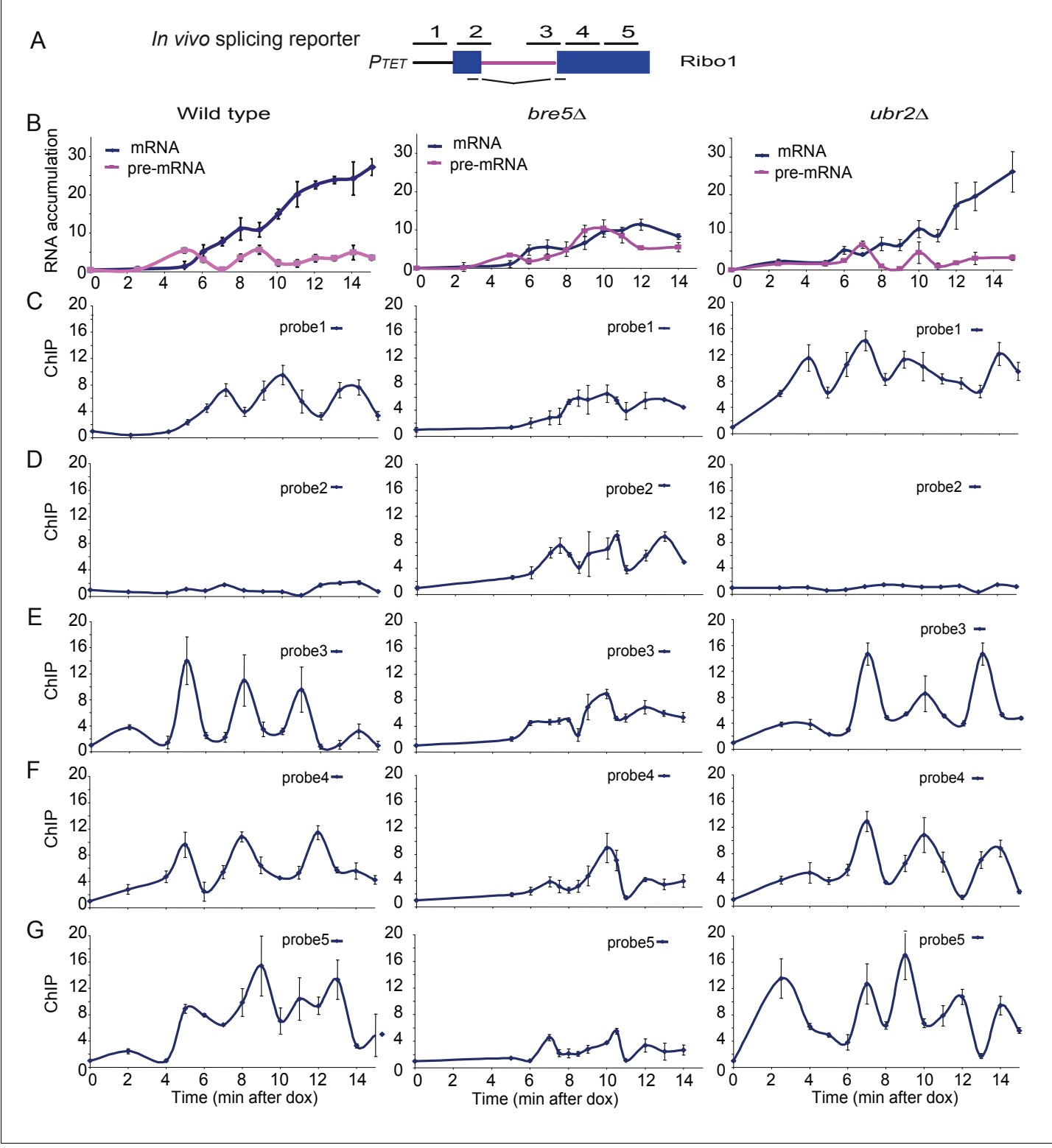

**Figure 2.** Analysis of ribosys reporter induction. (**A**) Diagram of theRibo1 gene. Exons are represented by rectangles, and the intron by a line with sequences at the ends. The reporter genes are based on previously described *ACT1-PGK1* constructs (*Alexander et al., 2010b*; *Hilleren and Parker, 2003*) expressed under control of a *tetO7-CYC1-UAS* promoter (*Bellí et al., 1998*). The lines above indicate amplicons analyzed in ChIP analyses: 1, 2, 3, 4, and 5 correspond to the promoter, the 5′SS, the 3′SS, the 5′ end of exon 2 and the 3′ end of exon 2, respectively. The lines below indicate the primers used in RT-qPCR to amplify the spliced RNA products. Unspliced pre-rRNA was detected using primer pair 3 over the 3′ SS. (**B**) RT-qPCR

*Figure 2 continued on next page*

*Figure 2 continued*

analysis of the accumulation of pre-mRNA and mRNA species during a time course (min) following doxycyclin addition (T0). Analyses were performed on the wild-type, *bre5Δ* and *ubr2Δ* strains as indicated. (C, D, E, F, G) ChIP analyses to detect RNAPII at the promoter (probe 1), the 5'SS (probe 2), the 3'SS (probe 3), the 5' end of exon 2 (probe 4) and the 3' end of exon 2 (probe 5). Data are presented as percentage of input relative to the uninduced level at T0. Cultures were as for B. All analyses were performed in biological triplicates. The qualitative results were consistent between replicates, however, slight differences in induction kinetics makes it difficult to combine these for statistical analyses. The graphs therefore show single experiments with technical replicates.

DOI: https://doi.org/10.7554/eLife.27082.006

The following source data and figure supplements are available for figure 2:

**Source data 1.** Source data for *Figure 2*.
DOI: https://doi.org/10.7554/eLife.27082.013
**Source data 2.** Replicates source data.
DOI: https://doi.org/10.7554/eLife.27082.014
**Figure supplement 1.** Effects of Bre5 loss on the Ribosys reporter is splicing dependent.
DOI: https://doi.org/10.7554/eLife.27082.007
**Figure supplement 1—source data 1.** ChIP1 source data.
DOI: https://doi.org/10.7554/eLife.27082.008
**Figure supplement 1—source data 2.** ChIP2 source data.
DOI: https://doi.org/10.7554/eLife.27082.009
**Figure supplement 1—source data 3.** ChIP3 source data.
DOI: https://doi.org/10.7554/eLife.27082.010
**Figure supplement 1—source data 4.** RT1 source data.
DOI: https://doi.org/10.7554/eLife.27082.011
**Figure supplement 1—source data 5.** RT2 and RT3 source data.
DOI: https://doi.org/10.7554/eLife.27082.012

Notably, these analyses reveal the initial time course of induction of an intron-containing transcript expressed from 'naïve' chromatin. In the wild-type strain, pre-mRNA was detected after 5 min, followed by the appearance of spliced mRNA at 6 min (*Figure 2B*). The mRNA levels then climbed progressively while pre-mRNA levels remained relatively constant. In the *bre5Δ* strain, pre-mRNA and mRNA were also detected after 5 min and 6 min, respectively. However, subsequent pre-mRNA levels were higher than in the wild-type, and mRNA failed to strongly accumulate. We conclude that splicing of the pre-mRNA reporter is strongly inhibited in the absence of Bre5.

In the wild-type, RNAPII levels show a striking periodicity following induction of the reporter (*Figure 2C–G*), as previously reported (*Alexander et al., 2010b*). In the *bre5Δ* strain, this periodicity was disrupted, with longer apparent pause durations. Notably, at the 5' splice site there is little clear pausing in the wild-type but much stronger signals in *bre5Δ* strains (Probe 2). In contrast, the wild-type RNAPII signals are high in the 3' region of exon2 (Probe 5), but much lower in *bre5Δ*, suggesting that RNAPII is delayed or lost prior to this region in the mutant. The accumulation of RNAPII in the vicinity of the 5' splice site (probe 2) was initially surprising, but there is a clear precedent in a previous report (*Chathoth et al., 2014*) that defects in pre-mRNA splicing lead to the accumulation of RNAPII at the 5' SS.

We interpret the data as showing that the oscillation in RNAPII association with the splicing reporter seen following initial induction is slower in the *bre5Δ* strain. Polymerase density increased in the 5' region of the transcription unit and decreased towards the 3' end. These findings would be consistent with a delay in the release of paused or slowed polymerases.

Notably, a reporter with a single nucleotide mutation at the 5' splice site that blocks splicing showed similar pre-mRNA, mRNA and RNAPII ChIP signals in wild-type and *bre5Δ* strains (*Figure 2—figure supplement 1*). This shows that the effects of Bre5 are specifically linked to pre-mRNA splicing. The putative ubiquitin ligase involved in the Bre5-dependent changes has not yet been identified. Ubr2 shows genetic and physical interactions with Bre5 and splicing factors (*Albers et al., 2003*; *Collins et al., 2007*; *Costanzo et al., 2010*). However, its absence did not clearly alter mRNA production or RNAPII elongation in the reporter construct (*Figure 2*).

## Loss of Bre5 affects splicing on endogenous genes

To test the relative effects of Bre5 on cotranscriptional versus post-transcriptional splicing of endogenous genes, we assessed the timing of splicing relative to pre-mRNA cleavage and polyadenylation (*Figure 3*). Unspliced but polyadenylated RNAs have failed to undergo cotranscriptional splicing and can be identified by RT-PCR. In strains lacking Bre5, the abundance of unspliced, poly(A)$^+$ RNA was decreased relative to spliced, poly(A)$^+$ RNA for two spliced transcripts tested (*RPL19B* and *RPL25*) (*Figure 3A*).

To determine whether the apparent effect of Bre5 loss on pre-mRNA splicing requires RNA binding, point mutations were generated in the RNA recognition motif that are predicted to impair binding activity without disrupting the structure of the protein ($K_{455}E$; $F_{435}A$; $Y_{421}A/F_{453}A$; $Y_{421}A/R_{423}E/F_{453}A$). The wild-type and mutant Bre5 constructs were expressed from the endogenous locus as C-terminal HTP fusions under the control of the *BRE5* promoter. To confirm that the point mutations impair functional RNA-binding, the wild-type and mutant forms of Bre5-HTP were compared by UV crosslinking in vivo. This was followed by purification on IgG, TEV elution, partial RNase digestion and nickel purification. Crosslinked RNAs were 5' labeled, and RNA-protein complexes were eluted and separated on an SDS gel. The recovery of Bre5 associated RNA was determined by autoradiography (*Figure 3B*, upper), while protein recovery was assessed by western blotting of the same gel (*Figure 3B*, lower). This showed that recovery of the mutant proteins was similar to wild-type Bre5, whereas each of the RRM mutations strongly reduced RNA association, with lowest binding for the $Y_{421}A/R_{423}E/F_{453}A$ triple mutant.

Relative to the strain expressing wild-type Bre5, the mutant strains showed reduced levels of unspliced, poly(A)$^+$ RNA for each of the transcripts tested (*RPL19B, ACT1* and *RPL25*), with the strongest effect in the $Y_{421}A/R_{423}E/F_{453}A$ triple mutant (*Figure 3C*).

We conclude that the loss of either Bre5 or its RNA-binding function, leads to a decrease in unspliced, polyadenylated RNA, indicating increased cotranscriptional splicing. Note that this assay shows steady state levels, and includes only transcripts on which RNAPII has reached the 3' end of the gene, whereas the RIBO1 reporter shows the effects on the kinetics of processing of nascent transcripts during initial induction.

## The catalytic subunit of RNAPII is ubiquitinated in the absence Bre5 or Ubp3

The association of Bre5 with RNAPII transcripts in the absence of DNA damage suggested that ubiquination of RNAPII subunits and deubiquitination by Bre5-Ubp3 occurs during normal growth. To assess this, all ubiquitinated proteins were purified using anti-ubiquitin antibodies, followed by western blotting for Rpo21 (*Figure 4A*). In an alternative approach, RNAPII was purified using antibodies against the Rpb3 subunit, followed by western blotting with anti-ubiquitin antibodies (*Figure 4B*). In both analyses, ubiquitinated Rpo21 was detected in the wild-type and was strongly elevated in cells lacking Ubp3, with a weaker increase in the absence of Bre5. In *Figure 4B*, the westerns were decorated with anti-Ub antibodies, which interact with free mono- and poly-ubiquitin and with all ubitiquitinated proteins in the input samples. Moreover, Bre5-Ubp3 have numerous targets in addition to RNAPII, many of which are increased in the mutant lysates.

The site of ubiquitination that was sensitive to Bre5-Ubp3 activity was identified by protein purification and mass-spectrometry (MS). To affinity purify RNAPII, we made use of a C-terminal fusion between Rpb3 and a tripartite tag containing calmodulin-binding peptide, a TEV-cleavage site and protein A (Rpb3-TAP). This was expressed from the chromosomal *Rpb3* locus under the control of the endogenous promoter and was fully functional with growth being indistinguishable from the wild type. Ubiquitinated RNAPII subunits were affinity purified from wild-type, *ubp3Δ* and *bre5Δ* strains by a His6-Ubiquitin expressed from the $P_{CUP1}$ promoter on a *CEN* plasmid, gel purified, subjected to in-gel trypsin digestion and analyzed by MS. This identified a number of sites that carried an additional lysine residue, derived from cleavage of attached ubiquitin molecules (*Figure 4C*). Ubiquitination of Rpo21 was detected at sites K452, K695 and K1246 in the wild-type strain. However, only ubiquitination of K1246 was detected in the *bre5Δ* strain. In the *ubp3Δ* strain, ubiquitination was detected at K695, and multiple peptides were recovered for K1246. Ubiquitination at K695 was not clearly altered in the *ubp3Δ* strain. In the *bre5Δ* strain, ubiquitination was detected only at K1246. A site of ubiquitination was also detected at Lys886 in Rbp2, the second largest RNAPII subunit, but

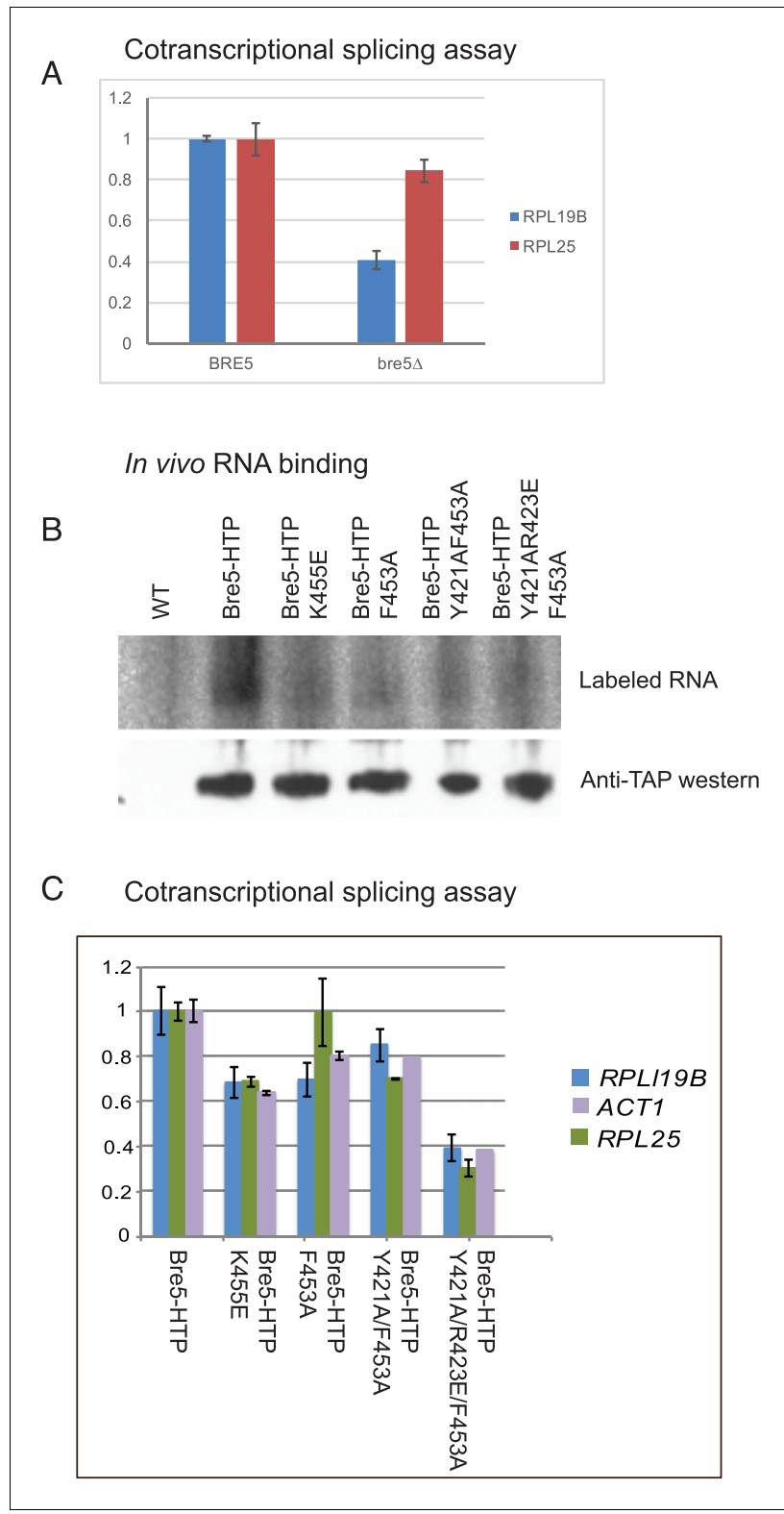

**Figure 3.** Strains lacking functional Bre5 show decreased co-transcriptional splicing. (**A**) RT-qPCR analysis of endogenous, polyadenylated and intron-containing transcripts using primer pairs over the 3' splice site expressed relative to non-intron-containing *PGI1* mRNA, in a wild type strain and in a strain lacking Bre5. The histogram shows the mean of three replicates with standard error, with the value in the *BRE5* strain set to 1. The reduced level of unspliced poly(A)$^+$ shows increased efficiency of cotranscriptional splicing associated with RNAPII that has

*Figure 3 continued on next page*

*Figure 3 continued*

reached the 3' end of the transcription unit. (B) Analysis of in-vivo RNA- binding activity of Bre5 with point mutants in the RRM. The top panel shows the recovery of radio-labelled RNA that was bound to Bre5 following in vivo crosslinking and multi-step, denaturing purification and separation by SDS-PAGE as described for CRAC analyses. The lower panel shows a western blot using an anti-TAP antibody against the tagged Bre5. (C) RT-qPCR analysis of endogenous unspliced, polyadenylated transcripts using primer pairs over the 3' splice site expressed relative to non-intron-containing *PGI1* mRNA. The histogram in a wild type HTP-tagged Bre5 strain and in strains carrying the point mutations in the RRM. The histogram shows the mean of three experiments with standard error, with the value in the strain expressing wild type Bre5 set to 1. The reduced level of unspliced, poly(A)$^+$ RNA in the mutant strains is interpreted as showing a requirement for a functional RRM in Bre5.

DOI: https://doi.org/10.7554/eLife.27082.015

The following source data is available for figure 3:

**Source data 1.** Source data for *Figure 3A*.
DOI: https://doi.org/10.7554/eLife.27082.016
**Source data 2.** Source data for *Figure 3C*.
DOI: https://doi.org/10.7554/eLife.27082.017

ubiquitination at this site was not clearly altered by the absence of either Bre5 or Ubp3. Ubiquitin with both K48 and K63 linkages was identified in the wild-type and *ubp3Δ* samples, making it unclear what linkage is present at K1246.

Multiple sites of ubiquitination were detected in RNAPII, and all should be enriched by purification. If ubiquitination is increased only at one site, this should be elevated relative to linear, unmodified peptides, which reflect the total recovery of Ub Rpo21. The intensities corresponding to modified peptides in each sample were normalized to the intensities of multiple, linear, non-modified peptides from Rpo21. Relative to total Ub Rpo21, the level of K1246 ubiquitination was estimated to be approximately five fold higher in *ubp3Δ* than the wild type, while the level of K695 was not clearly altered. This calculation is expected to under estimate the increased since the K1246 ubiquitination will contribute to total Ub Rpo21. Comparison to peptides from 'background' proteins indicated that the abundance of K1246 ubiquitination was approximately 10 fold elevated in the *ubp3Δ* strain.

To generate better quantitation by MS, we applied a targeted acquisition methodology - PRM (parallel reaction monitoring). PRM or SRM (selected reaction monitoring) are widely adopted strategies utilized to perform sensitive and reproducible quantitation of peptides including ubiquitinated peptides (*Gallien et al., 2014*). In brief, during acquisition, the mass spectrometer isolates only precursors corresponding to peptides of interest. The selected mass (precursor) is fragmented and obtained fragment ions are recorded. The intensity of selected fragment ions is then used for quantitation. For the analyses, Rpb3 was purified as described above, FASP digested and analyzed on an Orbitrap Fusion Lumos operated in PRM mode. Both modified (VVRPK*SLDAETEAEEDHMLK) and unmodified (VVRPK) peptides were readily detected in all samples containing Rpo21 co-purified with Rpb3-TAP (wild type, *bre5Δ* and *ubp3Δ*) and not in an untagged strain. Quantitation was performed on y10-y18 fragment ions by averaging intensity of the most intense ions between the replicas. These analyses indicated that levels of K1246-Ub in *bre5Δ* and *ubp3Δ* are 1.6 and 10.5 times higher, respectively, than in the wild type (*Figure 4D*). We conclude that loss of Bre5-Upb3 specifically increases ubiquination of Rpo21 at K1246.

The ubiquitination site at K1246 is located in an acid loop that is unstructured and therefore absent from published RNAPII crystal structures (*Figure 4E*) (*Sainsbury et al., 2013*). However, the position of the base of this loop (P1245 and E1253; residues shown as space filling in *Figure 4E*) strongly indicates that the site of ubiquitination lies close to the entrance to the active site of the polymerase. Ubiquitin in this position appears likely to clash with the entry of the DNA duplex into the active site (*Figure 4F*), and may also interfere with binding of transcription elongation factors, potentially explaining the basis for stalling of RNAPII by ubiquitination.

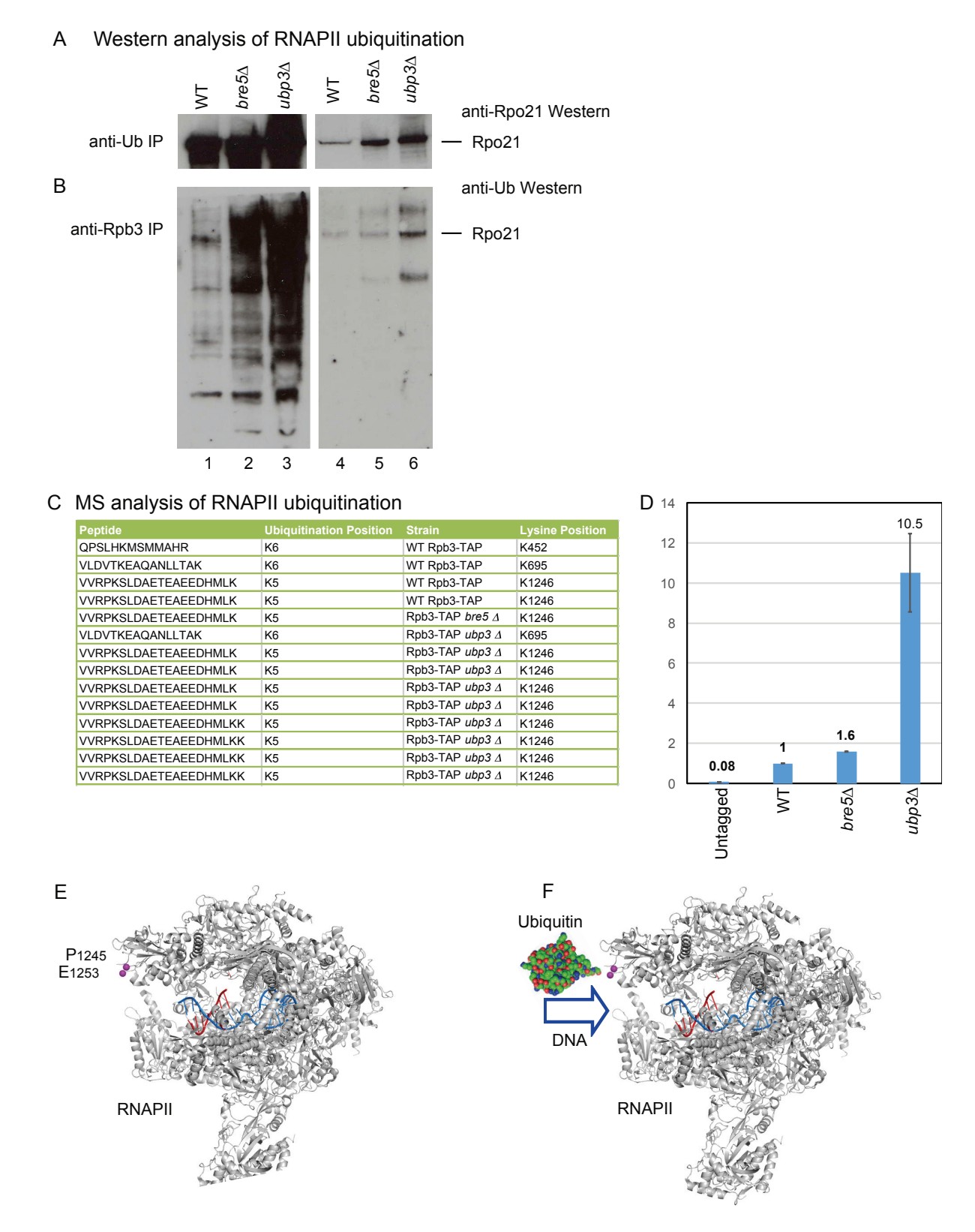

**Figure 4.** Identification of ubiquitination sites in RNAPII. (**A**) Analysis of Rpo21 immunoprecipitated from wild type cells, cells lacking Bre5 and cells lacking Ubp3 using an anti-ubiquitin antibody followed by Western blotting using an antibody against Rpo21. (**B**) Analysis of

*Figure 4 continued on next page*

*Figure 4 continued*

Rpo21 immunoprecipitated from wild type cells, cells lacking Bre5 and cells lacking Ubp3 using an anti-Rpb3 antibody followed by western blotting using an anti-ubiquitin antibody. (C) Table showing the ubiquitinated Rpo21 peptides from wild type cells, cells lacking Bre5 and cells lacking Ubp3, identified by mass spectrometry. (D) Quantitation of modified and unmodified peptides derived from the region surrounding Rpo21 K1246. (D) PRM-based quantitation of peptide harboring Rpo21 K1246-Ub. The sum of XICs (extracted ion chromatogram) corresponding to the four most intense ions was used to calculate the total area under curve and taken as a value representing recovery of Rpo21 K1246-Ub. The abundance of Rpo21 K1246-Ub in the wild type sample was used for normalization. Ratios of Rpo21 K1246-Ub identified in untagged, *bre5Δ* and *ubp3Δ* strains are shown relative to the wild type. (E) The structure of RNAPII was taken from PDB: 4bbs (*Sainsbury et al., 2013*) and kindly prepared by Altanta Cook (Edinburgh University). Protein residues are shown in gray, with the single stranded DNA template in red and the RNA transcript in blue. The ubiquitation site in Rpo21 at $K_{1246}$ is not visible in the crystal structure, but the neighboring sites at $P_{1245}$ and $E_{1253}$ are shown in magenta. (F) As panel D but with ubiquitin drawn in the approximate position of the crosslink and at a similar scale.

DOI: https://doi.org/10.7554/eLife.27082.018

The following source data and figure supplement are available for figure 4:

**Source data 1.** Source data for *Figure 4D*.
DOI: https://doi.org/10.7554/eLife.27082.020
**Figure supplement 1.** Quantitative MS analyses.
DOI: https://doi.org/10.7554/eLife.27082.019

## Ubiquitinated RNAPII is enriched over exon 2 of intron-containing genes

To map the locations of RNAPII containing ubiquitinated Rpo21, we applied modification CRAC (mCRAC) (*Figure 5A*) (*Milligan et al., 2016*). As for CRAC analyses, Rpo21-HTP was UV-crosslinked in actively growing cells and tandem affinity purified under denaturing conditions, prior to identification of associated RNAs by RT-PCR and illumina sequencing. This allows strand-specific mapping of the locations of modified RNAPII, with nucleotide resolution. During purification, ubiquitinated proteins were specifically enriched using a MultiDsk construct, comprised of five UBA domains from the yeast ubiquitin-binding protein Dsk2 fused to GST (*Wilson et al., 2012*). This binds ubiquitinated proteins with high affinity and specificity (*Figure 5—figure supplement 1A*). The autoradiograph of labeled, RNA bound Rpo21 purified by mCRAC (*Figure 5B*) shows increased recovery of Rpo21-Ub-RNA complexes in the *bre5Δ* strain. However, this was not clearly seen in all replicates.

The distribution of RNAPII is notably uneven along transcription units (*Churchman and Weissman, 2011*; *Mayer et al., 2015*; *Milligan et al., 2016*; *Nojima et al., 2015*). The level of ubiquitinated Rpo21 recovered via mCRAC was therefore mapped relative to total RNAPII from the same experiment. Alignment with the transcription start site (TSS) revealed strong depletion of RNAPII ubiquitination over the promoter proximal region of ~150 nt (*Figure 5D*). Conversely, alignment with the poly(A) site showed that the relative abundance of ubiquitinated RNAPII peaked ~200 nt upstream of the poly(A) site and then dropped sharply (*Figure 5E*). Alignment of intron-containing genes via the 3' SS showed a peak of ubiquitinated RNAPII at the 5' end of the exon 2 region (*Figure 5F*). Note that the relatively low number of intron-containing genes in budding yeast results in an uneven read distribution profile. In a *bre5Δ* strain the peaks of ubiquinated RNAPII were elevated immediately upstream of the poly(A) site (*Figure 5E*) and 3' SS (*Figure 5F*), and these differences are statistically significant (*Figure 5—figure supplement 1E–H*). Note, however, that the data are expressed in reads per million, so increased ubiquitination in the *bre5Δ* strain is not captured in the sequence data. Rather *Figure 5* and *Figure 5—figure supplement 1* show alterations in the relative density of Ub-Rpo21 along the genes. Notably, the distribution of ubiquitinated RNAPII is in overall agreement with the distribution of Bre5 (*Figure 1*) in CRAC analyses.

## Mutation of the ubiquitination site in Rpo21 decreases cotranscriptional splicing and alters RNAPII pausing

To test the effects of ubiquitin addition, a K1246R mutant form of Rpo21-HTP was generated and expressed from the chromosomal *RPO21* (*RPB1*) locus under the control of the endogenous promoter. Growth of the strain expressing only Rpo21$K_{1246}$R-HTP was not clearly different from the wild-type, showing the mutant protein to be functional.

Slowed transcription elongation by RNAPII will be associated with an increased signal in CRAC analyses, since more time is available for crosslinking to the transcript and the polymerase density

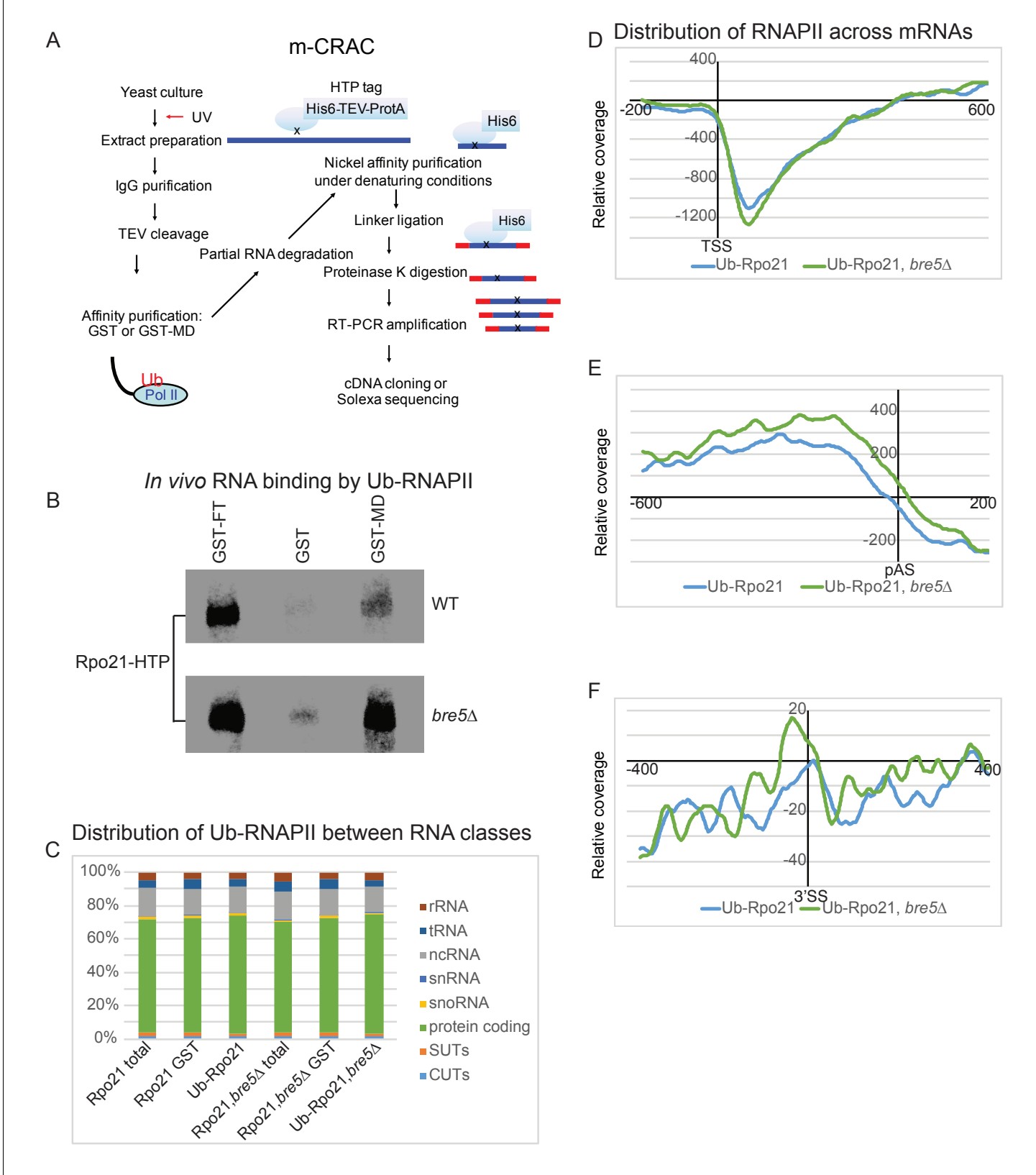

**Figure 5.** Transcriptome wide distribution of ubiquitinated Rpo21 by m-CRAC. (**A**) Outline of the m-CRAC procedure. Actively growing cells expressing Rpo21-HTP (His6 -TEV cleavage site – 2 copies of the Z domain of protein A) were UV-crosslinked at 254 nm for 1 min. Following lysis, Rpo21-HTP was
*Figure 5 continued on next page*

*Figure 5 continued*

affinity purified on an IgG column and released by TEV cleavage. Ubiquinated proteins were enriched on a GST-MultiDsk column (GST-MD) or mock enriched over GST alone. Associated RNA was partially denatured by RNase treatment, followed by protein denaturation by addition of guanidinium HCl to 4M. Denatured Rpo-21-His was bound to a nickel column in the same buffer. Following washing, bound RNAs were 5' labeled with [$^{32}$P], and linkers were added to both ends. Proteins were eluted with imidazole and separated by SDS-PAGE. Rpo21-RNA complexes were localized by autoradiography, excised from the gel and digested with proteinase K. RNAs released by this treatment were amplified by RT-PCR and identified by illumina sequencing. (**B**) Autoradiograph of labeled RNA bound to Rpo21 and purified by mCRAC. The top panel shows Rpo21 from wild type cells and the bottom panel from cells lacking Bre5. (**C**) Bar chart showing ubiquinated Rpo21 binding to different RNA classes. (**D**) Metagene analysis of the distribution of ubiquinated Rpo21 enrichment relative to total Rpo21, across protein-coding transcripts (5171 features) (*Xu et al., 2009*) aligned to the TSS in wild type cells (blue line) and cells lacking Bre5 (green line). The x axis shows the relative position in nt. The average of 2 m-CRAC replicates is shown, the standard deviation is shown in *Figure 5—figure supplement 1*. (**E**) As D, but transcripts were aligned to the pA sites. (**F**) Metagene analysis of the distribution of ubiquinated Rpo21 enrichment relative to total Rpo21, across intron-containing transcripts (288 features) relative to the 3'SS in wild type cells (blue line) and cells lacking Bre5 (green line). The x axis shows the relative position in nt. The average of 2 m-CRAC replicates is shown. The standard deviation is shown in *Figure 4—figure supplement 1*.

DOI: https://doi.org/10.7554/eLife.27082.021

The following figure supplement is available for figure 5:

**Figure supplement 1.** Analysis of transcriptome wide distribution of ubiquinated Rpo21 by mCRAC.

DOI: https://doi.org/10.7554/eLife.27082.022

will be higher. If ubiquitination of RNAPII is functionally linked to slowed elongation, the distribution of the non-ubiquitinated mutant K1246R, which is expected to reduce pausing, is predicted to be anti-correlated with RNAPII-Ub that is associated with increased pausing. This hypothesis was assessed in CRAC analyses (*Figure 6*).

The distribution Rpo21K$_{1246}$R was assessed across all protein-coding genes and is shown relative to the wild-type in *Figure 6A–C*. Close to the TSS and at the 3' splice site, the relative levels of Rpo21K$_{1296}$R RNAPII were reduced. Note, however, that the data maps the distribution of RNAPII across genes, so accumulation at any position results in apparent depletion elsewhere, and vice versa. At the poly(A) site, levels of Rpo21K$_{1296}$R showed a minimum ~200 nt upstream of the poly(A) site, but peaked immediately 3' to the p(A) site. Around the poly(A) site and at the 3'SS, the distribution of Rpo21K$_{1296}$R appeared to be reciprocal to that of Rpo21 ubiquitination, particularly in the *bre5Δ* background (*Figure 6A–C*). This supports the model that ubiquitination at K1246 is a physiologically important target of Bre5-Ubp3.

If ubiquitin-induced transcription pausing promotes cotranscriptional splicing, this is predicted to be reduced in strains expressing Rpo21K$_{1246}$R. Cotranscriptional splicing is expected to occur prior to 3' cleavage and polyadenylation, and this was assessed by determining the levels of unspliced pre-mRNA by qPCR on oligo(dT) primed RT products (*Figure 6D*). For three genes tested, *RPL29*, *RPL25* and *ACT1*, we saw increased levels of unspliced, poly(A)$^+$ pre-mRNA in two independent strains expressing Rpo21K$_{1246}$R.

Alterations in cotranscriptional splicing were also assessed transcriptome-wide in RNA-seq experiments performed on total (Ribominus) and poly(A)$^+$ selected RNA. For each individual intron-containing mRNA, reads mapping to introns and exons were quantified and the ratio intron/exon (I/E) was calculated and averaged between replicates. To compare transcripts from the strains expressing Rpo21-K$_{1246}$R and the wild-type Rpo21, we then calculated the log2((I/E)$_{Rpo21-K1246R}$/(I/E)$_{Rpo21}$). This value was positive in a metagene analysis of all intron-containing mRNAs (*Figure 6E*). Many spliced yeast genes are expressed at low levels during normal, vegetative growth. Analysis of all individual mRNAs showed increased levels of unspliced, poly(A)$^+$ RNA for all well-expressed genes (*Figure 6F*). This conclusion was further tested by considering only reads mapping to exon junctions (*Figure 6G*). The relative recovery of spliced mRNA versus unspliced pre-mRNA was expressed as the ratio of reads spanning exon–intron plus intron–exon to exon–exon junctions ((EI + IE)/2 EE) for each gene (*Milligan et al., 2016*; *Tuck and Tollervey, 2013*). Across all intron-containing genes, the unspliced to spliced ratio was higher for the Rpo21K$_{1246}$R mutant compared to the wild-type.

We also performed the same analyses on total RNA libraries depleted for rRNAs (Ribominus), which should allow the recovery of nascent transcripts (*Figure 6H–J*). Fewer reads mapping to mRNAs were recovered with this approach, but an increase in the level of unspliced pre-mRNA was found for almost all mRNAs with good read density. *Figure 6E–J* show the combined analysis of

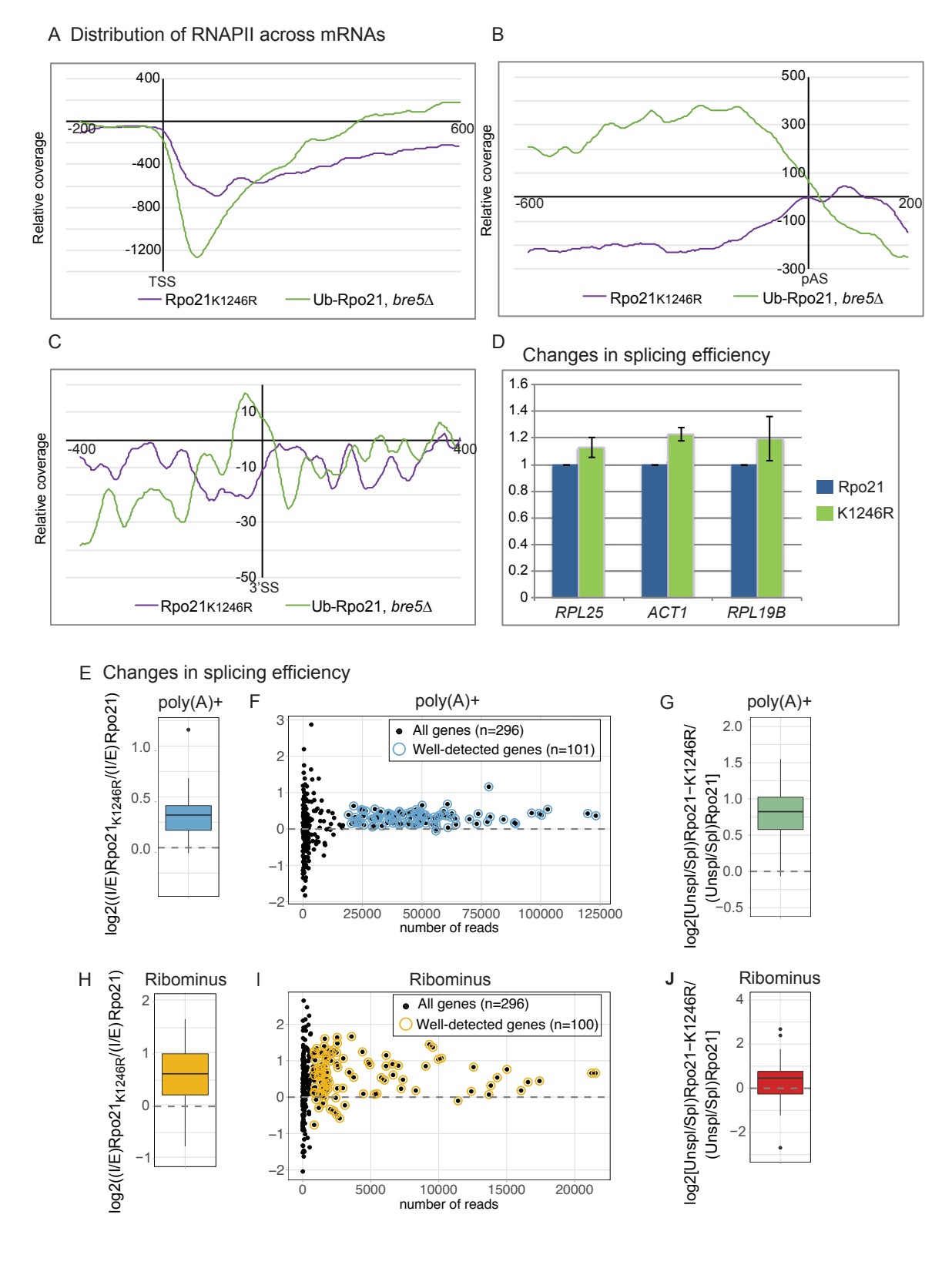

**Figure 6.** Loss of RNAPII ubiquitation is associated with reduced pausing and co-transcriptional splicing. (**A**) Metagene analysis of the distribution of Rpo21K1246R relative to wild-type Rpo21 (purple line) and ubiquitinated Rpo21 relative to total Rpo21 in strains lacking Bre5 (green line) across protein-

*Figure 6 continued on next page*

*Figure 6 continued*

coding transcripts relative to the TSS. The x axis shows the relative position in nt. The average of 2 replicates is shown. (**B**) As A, but transcripts were aligned to the pA sites. (**C**) Metagene analysis of the distribution of Rpo21K1246R relative to wild-type Rpo21 (purple line) and ubiquitinated Rpo21 relative to total Rpo21 in strains lacking Bre5 (green line) across intron-containing transcripts relative to the 3' SS. The x axis shows the relative position in nt. The average of 2 replicates is shown. (**D**) Quantitation of selected intron-containing transcripts by qPCR on oligo(dT) primed cDNAs, in a strain expressing Rpo21-HTP (blue bars) or Rpo21K1246R-HTP (green bars). The histogram shows the mean of three replicates with standard error. (**E**) Reads mapping to introns and exons were quantified in RNA-seq libraries prepared from poly(A)$^+$ RNA and the average intron/exon ratio (I/E) was calculated. The boxplot shows the distribution of log2((I/E)Rpo21-K1246R/(I/E)Rpo21) values for the top represented, intron-containing mRNAs (n = 101). The data indicate higher intron recovery in poly(A)$^+$ RNAs in the Rpo21-K1246R mutant. (**F**) For each intron-containing gene (n = 296), the log2((I/E)Rpo21-K$_{1246}$R/(I/E)Rpo21) from poly(A)$^+$ reads is plotted against the number of reads mapped to this gene. Almost all genes with enough coverage to be confidently analyzed (well-detected genes). (have a positive ratio, showing that cotranscriptional splicing efficiency was reduced in the Rpo21-K$_{1246}$R strain compared to wild-type. Confidently detected genes (n = 101) used for *Figure 6E* and *Figure 6—figure supplement 1A* are highlighted. (**G**) The relative recovery of unspliced pre-mRNA versus spliced mRNA (Unspl/Spl) was expressed as the ratio of reads spanning (exon-intron plus intron-exon) to exon_exon junctions in the poly(A)$^+$ RNA-seq libraries. For most mRNAs with sufficient coverage at the splice junctions (n = 85), the log2 ratio of (Unsp/Spl) of Rpo21-K1246R to Rpo21 was positive, indicating increased recovery of unspliced, polyadenylated pre-mRNAs. (**H**) As panel E, but using RNA depleted for rRNA (Ribominus) with confidently detected genes (n = 100). (**I**) As panel F, but using RNA depleted for rRNA (Ribominus). Confidently detected genes (n = 100) used for *Figure 6G* and *Figure 6—figure supplement 1B* are highlighted. (**J**) As panel (**G**). Relative recovery of unspliced versus spliced RNA (Unspl/Spl) was expressed as the ratio of reads spanning (exon_intron plus intron_exon) to exon_exon junctions recovered in total RNA depleted for rRNAs (n = 49).

DOI: https://doi.org/10.7554/eLife.27082.023

The following source data and figure supplement are available for figure 6:

**Source data 1.** Source data for *Figure 6D*.
DOI: https://doi.org/10.7554/eLife.27082.025
**Figure supplement 1.** Strains carrying a point mutation in the ubiquitinated LK1246 residue show loss of pausing at the 3' splice site and a reduction in co-transcriptional splicing.
DOI: https://doi.org/10.7554/eLife.27082.024

---

multiple datasets. Pairwise analyses of individual, replicate datasets lead to the same conclusions for both poly(A)$^+$ (*Figure 6—figure supplement 1A*) and ribominus samples (*Figure 6—figure supplement 1B*).

We conclude that single amino acid point mutation at the site of Bre5-Ubp3 sensitive RNAPII ubiquitination reduces the efficiency of cotranscriptional pre-mRNA splicing, transcriptome-wide.

## Discussion

The work presented here commenced with the, initially puzzling, identification of Bre5 and Ubp3 in synthetic-lethal screens with multiple RNA surveillance factors. We speculate that these negative interactions arise because RNAPII that is paused or stalled in response to ubiquitination has an increased tendency to release the nascent transcript. Accumulation of such aberrant RNA fragments in surveillance mutants may result in impaired growth.

Reproducible sites of ubiquitination were detected at positions K695 and K1246 in Rpo21 (Rpb1), the largest subunit of RNAPII and at K886 in Rpb2, the second largest subunit. However, only K1246 ubiquitination appeared to be clearly elevated in strains lacking Bre5 or Ubp3, indicating that this is the target of Bre5-Ubp3 activity. The increase in K1246 ubiquitination was more marked in the absence of the deubiquinase Ubp3 than its cofactor Bre5, perhaps indicating that Ubp3 can also function with other cofactors.

Sites of Rpo21 ubiquitination were previously identified at K330 and K695, based on the in vitro activity of the ubiquitin ligase Rsp5 (*Somesh et al., 2007*). K330 was shown to be important for the response to UV damage, but this was not the case for K695. However, a double mutation of Rpo21 K330R plus K695R was lethal, showing that these sites functionally overlap despite being far apart in the RNAPII structure (*Somesh et al., 2007*). Our analysis, in the absence of UV damage, identified ubiquitination at K1246 and K695, but not K330. This suggests the possibility that in RNAPII stalled by different mechanisms, splicing versus DNA damage, alternative ubiquitin ligases are recruited that lead to different primary ubiquitination sites, K1246 versus K330, respectively, with K695 providing an alternative site in each case. Based on genetic interaction data, we tested Ubr2 as a candidate E3 ligase, but saw no effects in the splicing reporter construct. A plausible alternative candidate is

the splicing factor Prp19, an E3 ligase that adds K63 ubiquitin chains to Prp3 during the spicing reaction (*Song et al., 2010*). However, the ubiquitin ligase activity of Prp19 is required for splicing to proceed, making it difficult to determine whether it is also involved in splicing-dependent transcription pausing. Prp3 is deubiquitinated by Usp4 (*Song et al., 2010*), and is therefore unlikely to be a target for Bre5-Ubp3.

The striking similarities between sites of Bre5 binding and elevated Ub-RNAPII, and the anti-correlated pattern in the Rpo21K1246R mutant, strongly support the model that this is the major, functionally relevant ubiquitination site under normal growth conditions. Bre5 binding and Ub-RNAPII were both depleted close to the TSS and elevated upstream from the poly(A) site and in exon 2. The preferential binding of Bre5 and RNAPII ubiquitination in exon 2 suggested a potential functional relationship to reports of splicing-induced transcription pausing in this region (*Alexander et al., 2010b*; *Carrillo Oesterreich et al., 2010*). Consistent with this, the absence of Bre5 or the loss of its RNA-binding activity had the effect of increasing co-transcriptional splicing efficiency, presumably as a consequence of increased pausing. Dramatic effects on a splicing reporter construct were seen in the absence of Bre5 following rapid induction, which were consistent with strongly enhanced transcription pausing. We postulate that during the initial rounds of cotranscriptional splicing on a naïve template, the system is more sensitive to perturbation.

These findings suggest the model that the normal function of splicing-linked RNAPII ubiquitination is to transiently slow or pause the polymerase to promote cotranscriptional splicing (*Figure 7*). To test this hypothesis, we expressed the ubiquitation-resistant Rpo21K1246R mutant, with the expectation that lack of ubiquitation would reduce transcriptional pausing and, hence, the efficiency of cotranscriptional splicing. This was shown to be the case for individual genes, which showed increased levels of pre-mRNAs that were unspliced but 3' cleaved and polyadenylated. RNA-seq analyses on total poly(A)$^+$ selected RNA confirmed this result transcriptome wide, when either all reads mapping to exons vs intron were compared, or splicing was more specifically assessed by comparing the ratio of sequence reads spanning exon–exon junction (spliced RNA) to exon–intron plus intron–exon junctions (unspliced). The finding that splicing was impaired by loss of the ubiquitination site in RNAPII, supports the model that efficient cotranscriptional splicing involves a ubiquitination/deubiquitination cycle.

It remains possible that the Rpo21K$_{1246}$R mutation impairs pre-mRNA splicing independent of alterations in transcription elongation. Previous analyses have identified important roles for RNAPII modification in promoting the recruitment of splicing factors. However, this involved the reversible phosphorylation of residues in the repetitive, regulatory C-terminal domain (CTD) of the RNAPII large subunit. In contrast, the K1246 ubiquitination site is distant from both the CTD and the RNA exit channel, making it less likely to directly interact with either the nascent transcript or the splicing machinery. Alternatively, the K1246R mutation might be postulated to alter the RNAPII elongation rate independent of RNA-processing events. The sequence data show only relative distribution of RNAPII along genes, not absolute levels, making changes in global elongation rates hard to assess. However, previously reported mutations that alter elongation rates map to the trigger loop, which is located in the enzymatic center of the polymerase distant from the position of K1246 (*Braberg et al., 2013*; *Kaplan et al., 2012*).

Given that the large majority of yeast genes lack introns, poly(A) site proximal changes in RNAPII occupancy and Bre5 binding are presumably generally independent of splicing. Conversely, elevated Bre5 occupancy at the 5' end of exon 2 appears to be independent of cleavage and polyadenylation, since it was observed even on genes with long exon 2 regions (*Figure 1G*). Previous reports of slowed RNAPII elongation on exon 2 regions have posited both splicing-dependent (*Alexander et al., 2010b*) and splicing-independent (*Carrillo Oesterreich et al., 2010*) mechanisms. It may be that both conclusions are correct, with different mechanisms operating at different locations. The mechanism by which ubiquitination might slow elongation has not been determined but, as indicated in *Figure 4F*, the position of the ubiquitin moiety on the catalytic component of RNAPII might be expected to interfere with entry of the DNA duplex into the active site. An alternative, non-exclusive, possibility might that ubiquitin addition interferes with back-tracking of stalled RNAPII complexes.

Major changes in RNAPII phosphorylation and apparent processivity take place close to the site of pre-mRNA 3' cleavage and polyadenylation (reviewed in [*Conaway and Conaway, 2015*; *Hsin and Manley, 2012*; *Porrua and Libri, 2015*]). We speculate that a surveillance or RNAPII

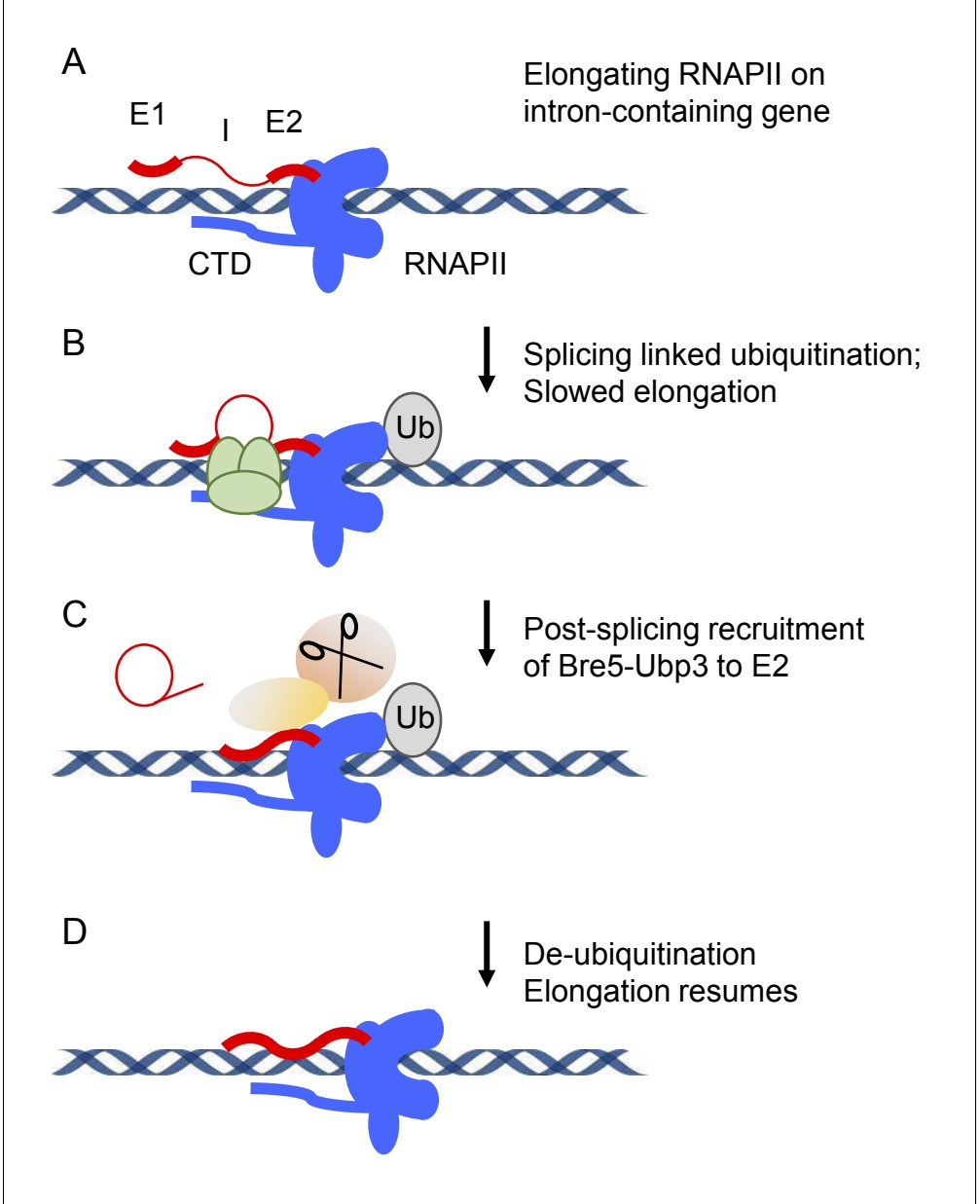

**Figure 7.** Model for splicing-linked RNAPII ubiquitination and de-ubiquitination. (**A**) Transcription on intron-containing gene. Exon 1 (E1), intron (I), exon2 (E2) and the C-terminal domain (CTD) of RNA polymerase II (RNAPII) are indicated. (**B**) Assembly of the spliceosome provokes RNAPII ubiquitination and slowed elongation. (**C**) Bre5-Ubp3 is recruited to the nascent transcripts following splicing. (**D**) De-ubiquitination by Ubp3 allows elongation to resume.

DOI: https://doi.org/10.7554/eLife.27082.026

remodeling step may occur upstream of the poly(A) site, as recently proposed for human cells (*Laitem et al., 2015*). This location is apparently associated with RNAPII ubiquitination as shown by the peaks of Ub-RNAPII and Bre5 binding at this position. RNAPII that is ubiquitination-resistant due to the $Rpo21K_{1296}R$ mutation showed a notably reduced density upstream of the poly(A) site relative to total RNAPII. This indicates that it may be less prone to pausing in this region and potentially avoids this surveillance activity. However, the $Rpo21K_{1296}R$ RNAPII shows increased occupancy downstream of the poly(A) site. We speculate that changes to RNAPII that would normally

accompany the pause upstream of the poly(A) site are absent or reduced in the Rpo21K$_{1296}$R mutant, leading to subsequent impairment of transcript release.

Finally, we note that Ubp3 was previously reported to interact with the chromatin-silencing factor Sir4, along with Sir2 and Sir3, while loss of Ubp3 was reported to increase the efficiency of gene silencing near telomeres and the inactive *HML* mating type locus (*Moazed and Johnson, 1996*). Based on our data, we hypothesize that this reflects increased RNAPII pausing or stalling in silenced genomic regions in the absence of Bre5-Ubp3-mediated deubiquitination. We also note that several recent reports link changes in alternative pre-mRNA splicing in human cells to RNAPII elongation rates (reviewed in [*Naftelberg et al., 2015*; *Saldi et al., 2016*]) and speculate that reversible ubiquitination of the transcribing polymerase may participate in regulating this activity.

# Materials and methods

## Strains and yeast culture
Standard procedures were used for the propagation and maintenance of yeast. All strains were constructed using a one-step PCR strategy. A full list of *Saccharomyces cerevisiae* strains used in this study can be found in *Supplementary file 1*, Table S3.

## RNA-binding assays using in vitro transcribed and translated proteins
Binding of [$^{35}$S]-labeled Bre5 and Ubp3 to homopolymeric nucleotides and biotinylated, in vitro-transcribed RNA was performed as previously described (*Milligan et al., 2008*; *Oeffinger et al., 2004*). Equivalent amounts of protein recovered from bound, unbound, and total fractions were separated by SDS-PAGE on a 4% to 12% Bis-Tris gel. Following migration, the gel was dried and protein was visualized by autoradiography.

## RNA-binding assays using HTP-tagged proteins purified from yeast
Cells lysis was performed essentially as described (*Granneman et al., 2011*). Bre5-HTP and Nab3-HTP were recovered from the lysate by binding to Dynabeads M-270 Epoxy (Invitrogen) coupled to rabbit IgG (Sigma) as described (*Fridy et al., 2014*). The bound protein was then resuspended in RNA-binding buffer (100 mM NaCl, 20 mM Tris (pH 7.4), 10% glycerol, 0.1 mM EDTA, 4 mM MgCl$_2$, 1 mM DTT, 40 µg of bovine serum albumin/ml, 200 µg of tRNA ml$^{-1}$). RNA binding was initiated by adding 2 pmol of radio-labeled RNA oligonucleotide to 3 pmol of bound protein and incubated at room temperature for 30 min. The unbound fraction was removed, the beads were washed and resuspended in RNA loading buffer. The unbound and bound fractions were analyzed on a 12% acrylamide, 8.3M Urea gel.

## CRAC
In vivo UV crosslinking at 254 nm in actively growing cells, RNA purification and cDNA generation were performed essentially as described (*Granneman et al., 2011*; *Schneider et al., 2012*). Following UV crosslinking, Bre5-HTP was recovered from the lysate by binding of the 2 copies of the Protein A tag to an IgG column and elution by TEV cleavage, essentially as described (*Granneman et al., 2011*; *Schneider et al., 2012*). RNA-protein complexes were subjected to partial RNase degradation, denatured by addition of Guanidinium HCl to 4M final concentration and bound to a nickel column via the His6 tag on Bre5. Linkers were added to the Bre5-associated RNAs on the nickel column and the bound protein-RNA complexes were eluted with imidazole and gel purified by SDS-PAGE (*Figure 1—figure supplement 1*). Following proteinase K digestion, RNAs were identified by reverse transcription and PCR amplification, followed by illumina HiSeq sequencing (performed by Edinburgh Genomics or Source Biosystems) or illumina miniSeq sequencing (performed in house).

## mCRAC analysis of ubiquitinated RNA PolII
In vivo UV crosslinking in actively growing cells, RNA purification and cDNA generation and were performed essentially as described (*Granneman et al., 2009*; *Schneider et al., 2012*). UV crosslinking at 254 nm was performed on actively growing cells, and Rpo21-HTP was recovered from the lysate by binding of the Protein A tag to an IgG column and elution by TEV cleavage, essentially as

described (*Granneman et al., 2009*; *Schneider et al., 2012*). Ubiquitinated Rpo21-HTP was separated using MultiDsk protein (*Wilson et al., 2012*). RNA-protein complexes were subjected to partial RNase degradation, denatured by addition of guanidinium HCl to 4M to denature and remove them from the MultiDsk column and bound to a nickel column via the His6 tag on Rpo21. Linkers were added to the Rpo21-associated RNAs on the nickel column and the bound protein-RNA complexes were eluted with imidazole and gel purified by SDS-PAGE. Following proteinase K digestion, RNAs were identified by reverse transcription and PCR amplification, followed by illumina sequencing. Illumina sequencing was performed by Edinburgh Genomics and Source Biosystems.

## Sequence data analysis for CRAC and mCRAC

Reads were pre-processed, aligned and analyzed as described (*Milligan et al., 2016*). The distribution of reads across transcript classes was determined using pyReadCounters from pyCRAC (*Webb et al., 2014*) and using genome annotation from Ensembl (EF4.74), supplemented with noncoding sequences as previously described (*Tuck and Tollervey, 2013*). For the metagene analysis, protein-coding genes and exon 2 contain 5171 and 288 features, respectively (*Xu et al., 2009*).

For Bre5 motif analysis, Bre5 reads were preprocessed using FLEXBAR (*Dodt et al., 2012*) with the parameters –at 1 –ao 4 and were filtered to retain only reads containing the 3′ adaptor. PCR duplicates were removed using pyFastqDuplicateRemover from pyCRAC (*Webb et al., 2014*). Reads were filtered to exclude low-complexity sequences (with more than 80% of one nucleotide) (*Tuck and Tollervey, 2013*). Reads were mapped to the yeast genome by using novoalign from Novocraft. PCR duplicates that were not collapsed during preprocessing due to sequencing errors or differential trimming at the 3′ end were collapsed (*Tuck and Tollervey, 2013*). After the exclusion of reads mapping to RNAPI and RNAPIII transcripts or originating from the mitochondrial genome (*Sayou et al., 2017*), reads mapping to protein-coding genes and containing single nucleotide deletions (indicating the precise nucleotide crosslinked to the bait protein) were selected using pyReadCounters from pyCRAC (*Webb et al., 2014*). To remove biases in sequence representation caused by abundantly expressed genes, overlapping reads were assembled into a single sequence using pyClusterReads from pyCRAC (*Webb et al., 2014*). A motif search was carried out using MEME (*Bailey et al., 2015*) on 2 different sets of 500 sequences for each dataset. A motif search was also done using the complete datasets using pyMotif from pyCRAC (*Webb et al., 2014*) and gave similar results. The 2 Bre5-HTP replicates were analyzed and gave a similar motif. Nab3 CRAC sequences were analyzed using the same pipeline and recovered the Nab3 motif (*Bresson et al., 2017*). CRAC sequences from Rpo21 (this study), the untagged BY4741 strain (this study), Set1 and Set2 (*Sayou et al., 2017*) were also analyzed as negative control and no motifs were found. CRAC sequences generated during this work have been deposited with GEO; accession number GSE94944, subseries GSE94941.

## Immunoprecipitation and western blotting

For analysis of the ubiquitination levels of Rpo21, cells were grown in YPD at 30°C to OD = 0.5 Whole cell lysates of cells were then prepared and immunoprecipitation was carried out with both an antibody against Rpb3 (1Y26, Neoclone, RRID:AB_1129174) and an anti-ubiquitin antibody (ab19247, Abcam, RRID:AB_444805). The immunoprecipitates were then separated by SDS-PAGE on a 4% to 12% Bis-Tris gel and transferred to nitrocellulose membrane (Hybond C, GE Lifesciences).

## RiboSys analysis

The RiboSys reporter used in this study was previously described (*Alexander et al., 2010a*). The RIBO1 reporter (*Figure 2A*) contains part of the *PGK1* open reading frame with the insertion of the *ACT1* pre-mRNA intron, integrated into the chromosomal *HIS3* locus under the control of an inducible tetO7 promoter. Cultures were pre-grown in synthetic dropout (SD) medium (Formedium) to $OD_{600nm}$ 0.5, transcription was induced by the addition of doxycycline to 4 µg/mL and aliquots were snap-frozen by pipetting into an equal volume of methanol sitting on dry ice. The frozen cells in the methanol slurry were pelleted by centrifugation and stored at −80°C.

## RNA extraction and RT-qPCR

RNA was extracted as previously described (*Tollervey, 1987*). Prior to the conversion to cDNA, 10 μg of total RNA was treated with Turbo-DNA free (Ambion) according to the manufacturer's protocol. For the Ribosys experiments cDNA was prepared from 5 μg of the DNase treated RNA using Retroscript (Ambion) with random decamers. cDNA was then diluted 1/20. qPCRs were performed in triplicate with Sensimix SYBR low-rox kit (Bioline) in a Stratagene MX3005P real-time PCR machine. Cycling parameters were 2 min at 94°C, then 50 cycles of 10 s at 94°C, 10 s at 63°C and 20 s at 72°C. For the endogenous transcripts, cDNA was prepared from 5 μg of the DNase treated RNA using Retroscript (Ambion) with oligodT. cDNA was then diluted 1/20. qPCRs were performed in triplicate with Sensimix SYBR low-rox kit (Bioline) in a Stratagene MX3005P real-time PCR machine. Cycling parameters were 10 min at 95°C, then 40 cycles of 15 s at 95°C, 15 s at 55°C and 15 s at 72°C.

## ChIP analysis

For ChIP analysis, 40 ml aliquots of culture were crosslinked for 10 min with 1% (v/v) formaldehyde and treated as described at http://www.ribosys.org/, using antibodies against Rpb3p (1Y26, Neoclone, RRID:AB_1129174). The DNA fragments (average size 350 bp) were amplified by qPCR using primers listed in *Supplementary file 1*, Table S4. ChIP data for the kinetic experiments are presented as percentage of input relative to uninduced level at T0. Experiments presented were performed at least in triplicate, with all qPCR assays also performed in triplicate. In each case, a representative experiment is shown.

## Protein expression and purification

MultiDsk was expressed and purified as previously described (*Wilson et al., 2012*). MultiDsk expression plasmid pGST-MD was propagated in BL-21 (DE3) cells. Transformed cells were grown overnight in starter culture in LB with 100 μg/ml ampicillin at 37°C. 600 ml of LB-Amp was inoculated with overnight starter culture and allowed to reach an optical density ($OD_{600}$) of 0.6. IPTG was added to a final concentration of 1 mM, and the culture was shifted to 30°C for 4 hr. Cells were harvested by centrifugation at 6000 g for 15 min, washed once in PBS, and frozen in liquid nitrogen. Cells were lysed and protein solubilized essentially as described (*Frangioni and Neel, 1993*). Briefly, thawed pellets were resuspended in STE buffer (10 mM Tris pH 8, 1 mM EDTA, 100 mM NaCl, 1 × Protease Inhibitor mix [284 ng/ml leupeptin, 1.37 μg/ml pepstatin A, 170 μg/ml phenylmethylsulfonyl fluoride and 330 μg/ml benzamindine]) with N-lauryl sarcosine added to a final concentration of 1.5%. After brief sonication and centrifugation, Triton X-100 was added to the supernatant to a final concentration of 3%, to mask the sarcosine. 0.8 ml of pre-equilibrated glutathione agarose beads (GE healthcare) were added to this solution, and the slurry incubated for 2–4 hr at 4°C. The beads were washed thoroughly in STE buffer containing 500 mM NaCl and 0.1% Triton X-100, followed by a 50 mM NaCl wash in the same buffer. The beads were then used directly for the mCRAC.

## RNA-seq

RNA was extracted as previously described (*Tollervey, 1987*). The RNA samples were either enriched for mRNA using NEBNext Oligo d(T)$_{25}$ (New England Biolabs) or depleted of rRNA using RiboMinus (Life Technologies) according to the manufacturer's instructions, producing poly(A)$^+$ or total RNA libraries, respectively. High throughput sequencing libraries were prepared using NEBNext Ultra Directional RNA Library Prep Kit for illumina according to the manufacturer's instructions. Illumina sequencing was carried out by Edinburgh Genomics. For each total RNA or polyA+ RNA libraries, 2 and 4 biological replicates were obtained for the strains expressing Rpo21-HTP (Rpo21) and Rpo21-K1246R-HTP (Rpo21-K1246R), respectively. Sequenced reads were reverse complemented using FASTX Toolkit, adaptors were trimmed and reads were quality filtered using Trimmomatic (*Bolger et al., 2014*). Reads were mapped to the yeast genome (*S. cerevisiae* genome version EF4.74, from Ensembl) using STAR (*Dobin et al., 2013*). To evaluate the splicing efficiency in the Rpo21-K1246R strain relative to the Rpo21 strain, reads mapping to introns (I) and exons (E) were counted in each dataset and for each intron-containing mRNA, using the Bedtools suite (*Quinlan and Hall, 2010*) and based on genome annotations from Ensembl (EF4.74). The ratio (I + 1)

/ (E + 1) (the pseudo-count of 1 avoids numerical instabilities) and the total count I + E were calculated for each transcript and averaged between replicates. The $\log2((I + 1/E + 1)_{Rpo21-K1246R}/(I + 1/E + 1)_{Rpo21})$ indicated whether a transcript was relatively more recovered as unspliced (positive value) or spliced (negative value) in the mutant compared to the wild-type. To select for transcripts with enough coverage, we used those with I + E above the mean (I + E) in both wild-type and mutant datasets in the polyA+ RNA libraries, resulting in a subset of 101 transcripts. We obtained a similar size subset for the total RNA libraries by selecting the 100 top transcripts based on I + E ranking. In a complementary approach, the relative abundance of unspliced and spliced reads was calculated as described (*Milligan et al., 2016*; *Tuck and Tollervey, 2013*). Briefly, reads were aligned to a database containing pre-mRNAs and mature mRNAs using Novoalign (Novocraft). The relative number of reads spanning the unspliced junctions (exon–intron, EI and intron–exon, IE) to spliced (exon-exon, EE) was calculated as (EI + IE)/2 EE for each transcript and averaged between replicates. The $\log2([(EI + IE)/2\ EE_{Rpo21-K1246R}] / [(EI + IE)/2\ EE_{Rpo21}])$ value was calculated for each transcript and indicated whether a transcript was relatively more recovered as unspliced (positive value) or spliced (negative value) in the mutant compared to the wild-type. RNA-seq sequences generated during this work have been deposited with GEO; accession number GSE94944, subseries GSE94942.

## Affinity purification of Rpb3-TAP for analytical mass spectrometry

Affinity purification of Rpb3-TAP was performed as previously (*Rigaut et al., 1999*) with the following modifications. Frozen yeast pellets, obtained from exponentially growing cells in synthetic medium minus tryptophan (optical density of 0.6 at 600 nm), after induction of His6-Ubiquitin with 0.1 mM CuSO4, were resuspended in TMN150 buffer (50 mM Tris-HCl pH 7.8, 150 mM NaCl, 1.5 mM MgCl$_2$, 0.1% NP-40, 5 mM beta-mercaptoethanol, 20 μM Lactacystin, 2 mM N-ethylmalemide). Clarified extract was incubated with IgG-Sepharose for 2 hr at 4°C. Washes and TEV cleavage were performed in TMN150 buffer. The ubiquitylated sub-units were purified via the His6-tagged ubiquitin on a Ni-NTA sepharose column, eluted and separated by SDS-PAGE and silver stained.

## SDS-PAGE analysis and in-gel digestion

The entire lane of the coomassie-stained gel was chopped into small pieces, destained and digested as described (*Shevchenko et al., 1996*). In brief, proteins were reduced in 10 mM dithiothreitol for 30 min at 37°C, alkylated in 55 mM iodoacetamide for 20 min at room temperature in the dark, and digested at 37°C overnight with 12.5 ng/μl trypsin (Proteomics; Grade; Sigma). Following digestion samples were then acidified with equal volume of 0.1% of trifluoroacetic acid and spun onto Stage-Tips (*Rappsilber et al., 2003*). Peptides were eluted in 20 μl 80% acetonitrile, 0.1% trifluoroacetic acid and were concentrated to 2 μl (concentrator 5301; Eppendorf AG, Hamburg, Germany). They were then diluted to 5 μl with 0.1% trifluoroacetic acid for liquid chromatography-tandem mass spectroscopy (LC-MS-MS) analysis.

## Affinity purification of Rpb3-TAP for quantitative mass spectrometry

Affinity purification of Rpb3-TAP was performed as previously (*Rigaut et al., 1999*) with the following modifications. Frozen yeast pellets from untagged and Rpb3-TAP, Rpb3TAP/ Δbre5 and Rpb3-TAP/Δubp3 cells, obtained from exponentially growing cells in YPDA (optical density of 0.6 at 600 nm), were resuspended in TMN150 buffer (50 mM Tris-HCl pH 7.8, 150 mM NaCl, 1.5 mM MgCl$_2$, 0.1% NP-40, 5 mM beta-mercaptoethanol, 20 μM Lactacystin, 2 mM N-ethylmalemide). Clarified extract was incubated with Dynabeads M-270 Epoxy (Invitrogen) coupled to rabbit IgG (Sigma) as described (ref Fridy et al. Nature Protocols 2014) for 1 hr at 4°C. Bound protein was denatured off the column in 0.1% Rapigest (Waters) for 30 min at 60°C. The samples were then prepared for Mass spectrometry by Filter-aided sample preparation (FASP) (*Wiśniewski et al., 2009*). In brief, proteins were reduced in 10 mM dithiothreitol for 30 min at 37°C, alkylated in 55 mM iodoacetamide for 20 min at room temperature in the dark, and digested at 37°C overnight with 12.5 ng/μl trypsin (Proteomics; Grade; Sigma). Following digestion, samples were then acidified with equal volume of 0.1% of trifluoroacetic acid and spun onto StageTips (*Rappsilber et al., 2003*). Peptides were eluted in 20 μl 80% acetonitrile, 0.1% trifluoroacetic acid and were concentrated to 2 μl (concentrator 5301; Eppendorf AG, Hamburg, Germany). They were then diluted to 5 μl with 0.1% trifluoroacetic acid for liquid

chromatography-tandem mass spectroscopy (LC-MS-MS) analysis. tandem mass spectroscopy (LC-MS-MS) analysis.

## MS analysis

Analytic MS analyses were performed on an LTQ-Orbitrap Velos mass spectrometer (Thermo Fisher Scientific, Bremen, Germany), coupled online to an Ultimate 3000 RSLCnano Systems (Dionex, Thermo Fisher Scientific, UK). To prepare an analytical column with a self-assembled particle frit (*Ishihama et al., 2002*), $C_{18}$ material (3 µm ReproSil-Pur C18-AQ; Maisch GmbH, Ammerbuch-Entringen, Germany) was packed into a spray emitter (75 µm inner diameter, 8 µm opening, 70 mm length; New Objectives, United States) by using an air pressure pump (Proxeon Biosystems, Odense, Denmark). Mobile phase A consisted of 0.1% formic acid in water and mobile phase B of 80% acetonitrile in0.1% formic acid. Peptides were loaded onto the column at a flow rate of 0.5 µL min$^{-1}$ and eluted at a flow rate of 0.2 µL min$^{-1}$ according to the following gradient: 2% to 40% mobile phase B in 150 min and then to 95% in 11 min. Mobile phase B was then reduced to 2% until 180 min. FTMS spectra were recorded at 60,000 resolution (scan range 300–1700 m/z) and the twenty most intense peaks with charge $\geq 2$ of the MS scan were selected with an isolation window of 2.0 Thomson in the ion trap for MS2 (normal scan, wideband activation, filling 5.0E5 ions for MS scan, 1.0E4 ions for MS2, maximum fill time 100 ms, dynamic exclusion for 60 s). Searches were conducted using the MASCOT search engine (Matrix Science) against the complete *Saccharomyces cerevisiae* protein database (*Saccharomyces* Genome Database, November 2009). MS accuracy was set to 6 ppm and MS/MS tolerance to ±0.6 Da. Trypsin was used as enzyme allowing two missed cleavages. Carbamidomethylation of cysteine was chosen as fixed modification, while oxidation of methionine and diglycine of lysine were chosen as variable modifications.

For quantitative MS, all targeted analyses were performed on high resolution/accurate mass Orbitrap Fusion Lumos instrument (Thermo) coupled to Ultimate3000 RSLC system fitted with EasySpray column (50 cm 2 µm particles). Targeted analyses were performed in scheduled tSIMtMS2 mode with a following parameters – 1.6 m/z isolation window, AGC target was set to 5e4 for both MS1 and MS2 and injection time was set to 70 ms. Retention time and m/z of targeted peptides were selected on the base of the prior experiments. Data analysis was performed in Skyline 3.7 (*MacLean et al., 2010*), using the following filtering settings: retention time was considered within 4 min of MS/MS ID, and mass tolerance at fragment ion level – 4ppm. The four most intense fragment ions were used for quantitation.

## Acknowledgements

We would like to thank Atlanta Cook for advice on mutagenesis of the Bre5 RRM and the RNAPII structure shown in *Figure 4*. This work was supported by Wellcome Trust funding to DT [109916], JB [104648], JR [108504], JEAR [093853], RA [200885] and G.K. [097383]. GK was also supported by the Medical Research Council. Work in the Wellcome Trust Centre for Cell Biology is supported by Wellcome Trust core funding [092076].

## Additional information

### Funding

| Funder | Grant reference number | Author |
| --- | --- | --- |
| Wellcome | 109916 | David Tollervey |
| Medical Research Council | | Grzegorz Kudla |
| Wellcome | 104648 | Jean D Beggs |
| Wellcome | 108504 | Juri Rappsilber |
| Wellcome | 093853 | Jane E A Reid |
| Wellcome | 097383 | Grzegorz Kudla |
| Wellcome | 092076 | David Tollervey |

| Wellcome | 200885 | Robin Allshire |
|----------|--------|----------------|

The funders had no role in study design, data collection and interpretation, or the decision to submit the work for publication.

## Author contributions

Laura Milligan, Conceptualization, Formal analysis, Validation, Investigation, Visualization, Writing—original draft, Writing—review and editing; Camille Sayou, Data curation, Software, Formal analysis, Validation, Investigation, Methodology, Writing—original draft, Writing—review and editing; Alex Tuck, Data curation, Software, Formal analysis, Investigation, Methodology, Writing—original draft, Writing—review and editing; Tatsiana Auchynnikava, Formal analysis, Methodology, Writing—review and editing; Jane EA Reid, Investigation, Methodology, Writing—review and editing; Ross Alexander, Flavia de Lima Alves, Investigation; Robin Allshire, Supervision, Funding acquisition, Project administration; Christos Spanos, Formal analysis, Investigation; Juri Rappsilber, Formal analysis, Supervision, Investigation, Methodology, Writing—review and editing; Jean D Beggs, Formal analysis, Supervision, Funding acquisition, Methodology, Project administration, Writing—review and editing; Grzegorz Kudla, Software, Investigation, Methodology, Writing—review and editing; David Tollervey, Conceptualization, Supervision, Funding acquisition, Writing—original draft, Project administration, Writing—review and editing

## Author ORCIDs

Camille Sayou (ID) http://orcid.org/0000-0002-8226-7272
David Tollervey (ID) http://orcid.org/0000-0003-2894-2772

## Decision letter and Author response

Decision letter https://doi.org/10.7554/eLife.27082.031
Author response https://doi.org/10.7554/eLife.27082.032

# Additional files

## Supplementary files

• Supplementary file 1. The file contains Supplementary Tables S1-S4.
DOI: https://doi.org/10.7554/eLife.27082.027

• Transparent reporting form
DOI: https://doi.org/10.7554/eLife.27082.028

## Major datasets

The following dataset was generated:

| Author(s) | Year | Dataset title | Dataset URL | Database, license, and accessibility information |
|-----------|------|---------------|-------------|--------------------------------------------------|
| Laura Milligan, Camille Sayou, Alex Tuck, Tatsiana Auchynnikava, Jane EA Reid, Ross Alexander, Flavia de Lima Alves, Robin Allshire, Christos Spanos, Juri Rappsilber, Jean D Beggs, Grzegorz Kudla, David Tollervey | 2017 | RNA polymerase II stalling at pre-mRNA splice sites is enforced by ubiquitination of the catalytic subunit | https://www.ncbi.nlm.nih.gov/geo/query/acc.cgi?acc=GSE94944 | Publicly available at the NCBI Gene Expression Omnibus (Accession no: GSE94944, subseries GSE94942) |

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
