## [Decision Letter]

Thank you for submitting your article "RNA polymerase II stalling at pre-mRNA splice sites is enforced by ubiquitination of the catalytic subunit" for consideration by *eLife*. Your article has been favorably evaluated by James Manley (Senior Editor) and three reviewers, one of whom is a member of our Board of Reviewing Editors. The reviewers have opted to remain anonymous.

The reviewers have discussed the reviews with one another and the Reviewing Editor has drafted this decision to help you prepare a revised submission.

This study describes the role of RNAPII ubiquitination in the regulation of polymerase pausing especially over the relatively rare introns in *S. cerevisiae* (yeast). The reviewers found the data presented to be potentially important and of significant interest in the field. However the paper is marred by some inadequate data description and missing analysis. With a thorough rewrite and additional controls this paper may be suitable for *eLife* publication. The key points that need better clarification or further work are as follows.

1) The paper needs a careful rewrite with especially a more thorough description of exactly what the figure data represents and how this has been obtained experimentally (both legends and text).

2) The in vitro RNA binding specificity of Bre5 (Figure 1) needs a more complete/convincing analysis. Other RNA substrates should be tested as well as purified Bre5 rather than IPs. As the data stands Bre5 RNA binding data looks very marginal.

3) CRAC meta analyses for Bre5 (Figure 1) need normalisation and statistics (confidence interval). Also specific genes (+/- introns) should be shown. At present this data could be interpreted as Bre5 binding only steady state mRNA not pre-mRNA. The CRAC data appears at odds with the calculations of Bre5 binding to unspliced/splice RNA in Supplementary file 1. Also in 1C why does untagged control give reads?

4) Figure 2 reporter data looks very nice. However the probe 2 data needs more discussion. Should comment on why generally there is a reduction in RNAPII ChIP oscillation with BRE ko except for probe 2 which gives an opposite effect. Clearly the loss of mRNA accumulation (panel B) is very significant.

5) Figure 3: These data are barely described (especially 3B and 3C). 3A appears to contradict Figure 2 CRAC data.

6) Inputs are needed in Figure 4. 4C is overexposed and not quantified. The mass. spec. data in Figure 4 needs quantification. Also the direct demonstration that K1246 (but not K1246R) is deubiquitinated by Ubp3-Bre5 is required.

7) Figure 5 data is hard to follow: mCRAC protocol is poorly explained especially the nature of GST-MD and MultiDsk. What is 5C showing? 5D-F lack statistical analysis (confidence intervals). Does this data match the Bre5 CRAC data (Figure 2)?

8) Figure 6: The rational for showing the Rpo21K1246R versus UbRpo21, bre5 ko is not explained. Why not simply compare with wt yeast strain?

---

## [Author Response]

This study describes the role of RNAPII ubiquitination in the regulation of polymerase pausing especially over the relatively rare introns in S. cerevisiae (yeast). The reviewers found the data presented to be potentially important and of significant interest in the field. However the paper is marred by some inadequate data description and missing analysis. With a thorough rewrite and additional controls this paper may be suitable for eLife publication. The key points that need better clarification or further work are as follows.1) The paper needs a careful rewrite with especially a more thorough description of exactly what the figure data represents and how this has been obtained experimentally (both legends and text).

We have gone through the text and figure legends in an attempt to more clearly explain the analyses used and data obtained.

2) The in vitro RNA binding specificity of Bre5 (Figure 1) needs a more complete/convincing analysis. Other RNA substrates should be tested as well as purified Bre5 rather than IPs. As the data stands Bre5 RNA binding data looks very marginal.

The in vitro binding assays that were initially shown used recombinant proteins rather than IPs.

To increase the relevance of the in vitro binding assays, we analyzed the crosslinking data for Bre5 and identified a preferred motif (UUUG). This motif was incorporated into oligonucleotides and shown to be preferentially bound by purified Bre5 (new Figure 1, Figure 1—figure supplement 1).

3) CRAC meta analyses for Bre5 (Figure 1) need normalisation and statistics (confidence interval).

The normalization is now better described in the revised text (subsection “Bre5 is an RNA binding protein that shows preferential association with exon 2 of spliced pre-mRNAs”, sixth paragraph) and confidence intervals are indicated in the revised version of Figure 1.

Also specific genes (+/- introns) should be shown. At present this data could be interpreted as Bre5 binding only steady state mRNA not pre-mRNA.

We include graphs showing the distribution of Bre5 on individual genes in the new Figure 2—figure supplement 1.

The CRAC data appears at odds with the calculations of Bre5 binding to unspliced/splice RNA in Supplementary file 1.

The CRAC data show preferential binding of Bre5 to Exon 2 and to spliced RNAs. We interpret this as indicating that Bre5-Ubp3 preferentially associate with the transcript following the successful completion of splicing, as indicated in the model (Figure 7). We have altered the text (subsection “Bre5 is an RNA binding protein that shows preferential association with exon 2 of spliced pre-mRNAs”, sixth paragraph) to make this clearer in the initial description of the data.

Also in 1C why does untagged control give reads?

The protocol includes an RT-PCR step. In consequence, amplification of the experimental background generates sequence reads in the negative control samples. This is always seen in CLIP and CRAC analyses. As noted in the text, the untagged control recovered ~15 fold fewer total reads than Bre5 following PCR amplification. We have altered the revised Figure 1 to indicate the relative recoveries, in numbers of mapped reads, above the bar graphs.

4) Figure 2 reporter data looks very nice. However the probe 2 data needs more discussion.

The accumulation of RNAPII in the vicinity of the 5’ splice site (probe 2) was initially surprising, but there is a clear precedent for this. Work by the Beggs lab (Chathoth et al., 2014) showed that other defects in pre-mRNA splicing lead to the accumulation of RNAPII at the 5’ SS. We discuss this in the revised text (subsection “Loss of Bre5 results in RNAPII stalling and decreased mRNA on an inducible reporter”, fourth paragraph).

Should comment on why generally there is a reduction in RNAPII ChIP oscillation with BRE ko except for probe 2 which gives an opposite effect. Clearly the loss of mRNA accumulation (panel B) is very significant.

We interpret the data as showing that the oscillation is slower but covers a greater region of the transcription unit. This would be consistent with a delay in the release of the paused polymerases. We have altered the text (subsection “Loss of Bre5 results in RNAPII stalling and decreased mRNA on an inducible reporter”, fifth paragraph) to include this interpretation of the data.

5) Figure 3: These data are barely described (especially 3B and 3C).

We have included a more complete description of these results in the revised text (subsection “Loss of Bre5 affects splicing on endogenous genes”).

3A appears to contradict Figure 2 CRAC data.

No CRAC data are presented in Figure 2. The comment may refer to the RT-PCR data in Figure 2. If so, we would point out that Figure 3 shows the analysis of oligo(dT) primed sequence data. In consequence, loss of splicing is not seen if RNAPII is not released. We have altered the text to make this clearer (subsection “Loss of Bre5 affects splicing on endogenous genes”).

6) Inputs are needed in Figure 4.

We now show the inputs for Figure 4.

4C is overexposed and not quantified.

We have removed this figure from the revised manuscript. The figure showed enrichment of ubiquitinated Rpo21, purified via the expression of Ub-His6. However, Ub-His is expressed under the control of a P_CUP_ promoter, whereas endogenous ubiquitin is largely generated from fusions with ribosomal proteins. In consequence, the fraction of Ub that is expected to be expressed with the tag is low, lead to poor signal to noise ratio.

In the revised text, we point out that data in Figure 5 show an increase in RNA recovery with Rpo21-Ub in the *bre5∆* strain (subsection “Ubiquinated RNAPII is enriched over exon 2 of intron-containing genes”, first paragraph). This is consistent with the prediction that ubiquitation of the functional, transcriptionally engaged RNAPII population is increased in the strain lacking Bre5.

The mass. spec. data in Figure 4 needs quantification.

In order to perform quantitation of modified peptides we utilized two strategies. First, we estimated 5-fold relative enrichment of K1246 in *ubp3Δ* background normalized to several peptides representing global Rpo21 ubiquitination. This estimate is based on Ub-his enriched data, and therefore reflects levels of ubiquitination at K1246 relative to other all sites ubiquitination. This clearly indicated that only K1246 was elevated in the *ubp3∆* strain. In addition, we applied targeted acquisition (PRM) to assay level of K1246-Ub modification in affinity purified Rpo21-Rpb3TAP. Targeted methodologies such as PRM or SRM are utilized for precise and sensitive quantitation of K-GG modified peptides. The modified peptide containing K1246-Ub was readily detected by PRM experiment in all strains containing Rpo21 and not in untagged strain as expected. We quantified intensity of y13-y19 fragment ions originating from the modified peptide in two independent biological replicas and conclusively established that level of Rpo21 K1246-Ub is 1.6 fold higher in *bre5∆* and at least 10.5 fold higher in *upb3∆* than in the wild type. Three additional authors have been included in respect to these data.

Also the direct demonstration that K1246 (but not K1246R) is deubiquitinated by Ubp3-Bre5 is required.

Our understanding is that reproducing site-specific deubiquitination in vitro is very difficult. Treatment of RNAPII with Ubp3-Bre5 is very likely to result in deubiquitination at all sites. Establishing a specific deubiquitination system would be very useful for future analyses, but we feel that this lies beyond the scope of the current manuscript. Clearly the K1246R site cannot be deubiquitinated.

7) Figure 5 data is hard to follow: mCRAC protocol is poorly explained especially the nature of GST-MD and MultiDsk.

We have modified the text (subsection “Ubiquinated RNAPII is enriched over exon 2 of intron-containing genes”) and the legend to Figure 5 to make these points clearer.

What is 5C showing?

This panel was of limited use and for clarity it has been omitted from the revised manuscript.

5D-F lack statistical analysis (confidence intervals).

Confidence intervals are now shown in Figure 5—figure supplement 1.

Does this data match the Bre5 CRAC data (Figure 2)?

As noted above, CRAC data are not shown in Figure 2. The comment may refer to Figure 1. If so, there is general agreement between the distribution of Bre5 and Ub-RNAPII. We have altered the text to make the clearer (subsection “Ubiquinated RNAPII is enriched over exon 2 of intron-containing genes”).

8) Figure 6: The rational for showing the Rpo21K1246R versus UbRpo21, bre5 ko is not explained. Why not simply compare with wt yeast strain?

The Rpo21K1246R data are expressed relative to wild-type Rpo21. We make this clear in the revised text and figure legend. We show these data together with Ub-Rpo21 because, if Ub-RNAPII is associated with slowed elongation, we would expect its distribution to be anti-correlated with the non-Ub mutant K1246R. This was indeed seen for both 3’SS and poly(A) proximal regions. We have altered the text (subsection “Ubiquinated RNAPII is enriched over exon 2 of intron-containing genes”) to better explain this important point.

---

## [Decision Letter]

Thank you for submitting your article "RNA polymerase II stalling at pre-mRNA splice sites is enforced by ubiquitination of the catalytic subunit" for consideration by *eLife*. Your article has been favorably evaluated by James Manley (Senior Editor) and three reviewers, one of whom is a member of our Board of Reviewing Editors. The reviewers have opted to remain anonymous.

The reviewers have discussed the reviews with one another and the Reviewing Editor has drafted this decision to help you prepare a revised submission.

This study describes the role of RNAPII ubiquitination in the regulation of polymerase pausing especially over the relatively rare introns in *S. cerevisiae* (yeast). The reviewers found the data presented to be potentially important and of significant interest in the field. However the paper is marred by some inadequate data description and missing analysis. With a thorough rewrite and additional controls this paper may be suitable for *eLife* publication. The key points that need better clarification or further work are as follows.

1) The paper needs a careful rewrite with especially a more thorough description of exactly what the figure data represents and how this has been obtained experimentally (both legends and text).

2) The in vitro RNA binding specificity of Bre5 (Figure 1) needs a more complete/convincing analysis. Other RNA substrates should be tested as well as purified Bre5 rather than IPs. As the data stands Bre5 RNA binding data looks very marginal.

3) CRAC meta analyses for Bre5 (Figure 1) need normalisation and statistics (confidence interval). Also specific genes (+/- introns) should be shown. At present this data could be interpreted as Bre5 binding only steady state mRNA not pre-mRNA. The CRAC data appears at odds with the calculations of Bre5 binding to unspliced/splice RNA in Supplementary file 1. Also in 1C why does untagged control give reads?

4) Figure 2 reporter data looks very nice. However the probe 2 data needs more discussion. Should comment on why generally there is a reduction in RNAPII ChIP oscillation with BRE ko except for probe 2 which gives an opposite effect. Clearly the loss of mRNA accumulation (panel B) is very significant.

5) Figure 3: These data are barely described (especially 3B and 3C). 3A appears to contradict Figure 2 CRAC data.

6) Inputs are needed in Figure 4. 4C is overexposed and not quantified. The mass. spec. data in Figure 4 needs quantification. Also the direct demonstration that K1246 (but not K1246R) is deubiquitinated by Ubp3-Bre5 is required.

7) Figure 5 data is hard to follow: mCRAC protocol is poorly explained especially the nature of GST-MD and MultiDsk. What is 5C showing? 5D-F lack statistical analysis (confidence intervals). Does this data match the Bre5 CRAC data (Figure 2)?

8) Figure 6: The rational for showing the Rpo21K1246R versus UbRpo21, bre5 ko is not explained. Why not simply compare with wt yeast strain?

---

## [Author Response]

This study describes the role of RNAPII ubiquitination in the regulation of polymerase pausing especially over the relatively rare introns in S. cerevisiae (yeast). The reviewers found the data presented to be potentially important and of significant interest in the field. However the paper is marred by some inadequate data description and missing analysis. With a thorough rewrite and additional controls this paper may be suitable for eLife publication. The key points that need better clarification or further work are as follows.1) The paper needs a careful rewrite with especially a more thorough description of exactly what the figure data represents and how this has been obtained experimentally (both legends and text).

We have gone through the text and figure legends in an attempt to more clearly explain the analyses used and data obtained.

2) The in vitro RNA binding specificity of Bre5 (Figure 1) needs a more complete/convincing analysis. Other RNA substrates should be tested as well as purified Bre5 rather than IPs. As the data stands Bre5 RNA binding data looks very marginal.

The in vitro binding assays that were initially shown used recombinant proteins rather than IPs.

To increase the relevance of the in vitro binding assays, we analyzed the crosslinking data for Bre5 and identified a preferred motif (UUUG). This motif was incorporated into oligonucleotides and shown to be preferentially bound by purified Bre5 (new Figure 1, Figure 1—figure supplement 1).

3) CRAC meta analyses for Bre5 (Figure 1) need normalisation and statistics (confidence interval).

The normalization is now better described in the revised text (subsection “Bre5 is an RNA binding protein that shows preferential association with exon 2 of spliced pre-mRNAs”, sixth paragraph) and confidence intervals are indicated in the revised version of Figure 1.

Also specific genes (+/- introns) should be shown. At present this data could be interpreted as Bre5 binding only steady state mRNA not pre-mRNA.

We include graphs showing the distribution of Bre5 on individual genes in the new Figure 2—figure supplement 1.

The CRAC data appears at odds with the calculations of Bre5 binding to unspliced/splice RNA in Supplementary file 1.

The CRAC data show preferential binding of Bre5 to Exon 2 and to spliced RNAs. We interpret this as indicating that Bre5-Ubp3 preferentially associate with the transcript following the successful completion of splicing, as indicated in the model (Figure 7). We have altered the text (subsection “Bre5 is an RNA binding protein that shows preferential association with exon 2 of spliced pre-mRNAs”, sixth paragraph) to make this clearer in the initial description of the data.

Also in 1C why does untagged control give reads?

The protocol includes an RT-PCR step. In consequence, amplification of the experimental background generates sequence reads in the negative control samples. This is always seen in CLIP and CRAC analyses. As noted in the text, the untagged control recovered ~15 fold fewer total reads than Bre5 following PCR amplification. We have altered the revised Figure 1 to indicate the relative recoveries, in numbers of mapped reads, above the bar graphs.

4) Figure 2 reporter data looks very nice. However the probe 2 data needs more discussion.

The accumulation of RNAPII in the vicinity of the 5’ splice site (probe 2) was initially surprising, but there is a clear precedent for this. Work by the Beggs lab (Chathoth et al., 2014) showed that other defects in pre-mRNA splicing lead to the accumulation of RNAPII at the 5’ SS. We discuss this in the revised text (subsection “Loss of Bre5 results in RNAPII stalling and decreased mRNA on an inducible reporter”, fourth paragraph).

Should comment on why generally there is a reduction in RNAPII ChIP oscillation with BRE ko except for probe 2 which gives an opposite effect. Clearly the loss of mRNA accumulation (panel B) is very significant.

We interpret the data as showing that the oscillation is slower but covers a greater region of the transcription unit. This would be consistent with a delay in the release of the paused polymerases. We have altered the text (subsection “Loss of Bre5 results in RNAPII stalling and decreased mRNA on an inducible reporter”, fifth paragraph) to include this interpretation of the data.

5) Figure 3: These data are barely described (especially 3B and 3C).

We have included a more complete description of these results in the revised text (subsection “Loss of Bre5 affects splicing on endogenous genes”).

3A appears to contradict Figure 2 CRAC data.

No CRAC data are presented in Figure 2. The comment may refer to the RT-PCR data in Figure 2. If so, we would point out that Figure 3 shows the analysis of oligo(dT) primed sequence data. In consequence, loss of splicing is not seen if RNAPII is not released. We have altered the text to make this clearer (subsection “Loss of Bre5 affects splicing on endogenous genes”).

6) Inputs are needed in Figure 4.

We now show the inputs for Figure 4.

4C is overexposed and not quantified.

We have removed this figure from the revised manuscript. The figure showed enrichment of ubiquitinated Rpo21, purified via the expression of Ub-His6. However, Ub-His is expressed under the control of a P_CUP_ promoter, whereas endogenous ubiquitin is largely generated from fusions with ribosomal proteins. In consequence, the fraction of Ub that is expected to be expressed with the tag is low, lead to poor signal to noise ratio.

In the revised text, we point out that data in Figure 5 show an increase in RNA recovery with Rpo21-Ub in the *bre5∆* strain (subsection “Ubiquinated RNAPII is enriched over exon 2 of intron-containing genes”, first paragraph). This is consistent with the prediction that ubiquitation of the functional, transcriptionally engaged RNAPII population is increased in the strain lacking Bre5.

The mass. spec. data in Figure 4 needs quantification.

In order to perform quantitation of modified peptides we utilized two strategies. First, we estimated 5-fold relative enrichment of K1246 in *ubp3Δ* background normalized to several peptides representing global Rpo21 ubiquitination. This estimate is based on Ub-his enriched data, and therefore reflects levels of ubiquitination at K1246 relative to other all sites ubiquitination. This clearly indicated that only K1246 was elevated in the *ubp3∆* strain. In addition, we applied targeted acquisition (PRM) to assay level of K1246-Ub modification in affinity purified Rpo21-Rpb3TAP. Targeted methodologies such as PRM or SRM are utilized for precise and sensitive quantitation of K-GG modified peptides. The modified peptide containing K1246-Ub was readily detected by PRM experiment in all strains containing Rpo21 and not in untagged strain as expected. We quantified intensity of y13-y19 fragment ions originating from the modified peptide in two independent biological replicas and conclusively established that level of Rpo21 K1246-Ub is 1.6 fold higher in *bre5∆* and at least 10.5 fold higher in *upb3∆* than in the wild type. Three additional authors have been included in respect to these data.

Also the direct demonstration that K1246 (but not K1246R) is deubiquitinated by Ubp3-Bre5 is required.

Our understanding is that reproducing site-specific deubiquitination in vitro is very difficult. Treatment of RNAPII with Ubp3-Bre5 is very likely to result in deubiquitination at all sites. Establishing a specific deubiquitination system would be very useful for future analyses, but we feel that this lies beyond the scope of the current manuscript. Clearly the K1246R site cannot be deubiquitinated.

7) Figure 5 data is hard to follow: mCRAC protocol is poorly explained especially the nature of GST-MD and MultiDsk.

We have modified the text (subsection “Ubiquinated RNAPII is enriched over exon 2 of intron-containing genes”) and the legend to Figure 5 to make these points clearer.

What is 5C showing?

This panel was of limited use and for clarity it has been omitted from the revised manuscript.

5D-F lack statistical analysis (confidence intervals).

Confidence intervals are now shown in Figure 5—figure supplement 1.

Does this data match the Bre5 CRAC data (Figure 2)?

As noted above, CRAC data are not shown in Figure 2. The comment may refer to Figure 1. If so, there is general agreement between the distribution of Bre5 and Ub-RNAPII. We have altered the text to make the clearer (subsection “Ubiquinated RNAPII is enriched over exon 2 of intron-containing genes”).

8) Figure 6: The rational for showing the Rpo21K1246R versus UbRpo21, bre5 ko is not explained. Why not simply compare with wt yeast strain?

The Rpo21K1246R data are expressed relative to wild-type Rpo21. We make this clear in the revised text and figure legend. We show these data together with Ub-Rpo21 because, if Ub-RNAPII is associated with slowed elongation, we would expect its distribution to be anti-correlated with the non-Ub mutant K1246R. This was indeed seen for both 3’SS and poly(A) proximal regions. We have altered the text (subsection “Ubiquinated RNAPII is enriched over exon 2 of intron-containing genes”) to better explain this important point.